# Distinguishing different modes of growth using single-cell data

**Prathitha Kar[1,2], Sriram Tiruvadi-Krishnan[3], Jaana Männik[3], Jaan Männik[3]\*, Ariel Amir[1]\***

[1]Harvard John A. Paulson School of Engineering and Applied Sciences, Harvard University, Cambridge, United States; [2]Department of Chemistry and Chemical Biology, Harvard University, Cambridge, United States; [3]Department of Physics and Astronomy, University of Tennessee, Knoxville, United States

**Abstract** Collection of high-throughput data has become prevalent in biology. Large datasets allow the use of statistical constructs such as binning and linear regression to quantify relationships between variables and hypothesize underlying biological mechanisms based on it. We discuss several such examples in relation to single-cell data and cellular growth. In particular, we show instances where what appears to be ordinary use of these statistical methods leads to incorrect conclusions such as growth being non-exponential as opposed to exponential and vice versa. We propose that the data analysis and its interpretation should be done in the context of a generative model, if possible. In this way, the statistical methods can be validated either analytically or against synthetic data generated via the use of the model, leading to a consistent method for inferring biological mechanisms from data. On applying the validated methods of data analysis to infer cellular growth on our experimental data, we find the growth of length in *E. coli* to be non-exponential. Our analysis shows that in the later stages of the cell cycle the growth rate is faster than exponential.

\*For correspondence:
jmannik@utk.edu (JM);
arielamir@seas.harvard.edu (AA)

**Competing interest:** The authors declare that no competing interests exist.

## Editor's evaluation

In this manuscript, the authors describe a generative model-based framework to better analyze stochastic growth data, including bacterial cell growth. They show how this framework can be applied to gain insight into the processes underlying these phenomena. This work is well-supported by simulations and data analysis and will likely be of interest to those trying to understand the processes governing bacterial growth, as well as those studying stochastic growth processes in biology more broadly.

## Introduction

The last decade has seen a tremendous increase in the availability of high-quality large datasets in biology, in particular in the context of single-cell level measurements. Such data are complementary to 'bulk' measurements made over a population of cells. They have led to new biological paradigms and motivated the development of quantitative models (*Osella et al., 2017*; *Facchetti et al., 2017*; *Ho et al., 2018*; *Soifer et al., 2016*; *Jun et al., 2018*; *Amir and Balaban, 2018*; *Kohram et al., 2021*). Nevertheless, they have also led to new challenges in data analysis, and here we will point out some of the pitfalls that exist in handling such data. In particular, we will show that the commonly used procedure of binning data and linear regression may hint at specific functional relations between the two variables plotted that are inconsistent with the true functional relations. As we shall show, this may come about due to the 'hidden' noise sources that affect the binning procedure and the

**eLife digest** All cells – from bacteria to humans – tightly control their size as they grow and divide. Cells can also change the speed at which they grow, and the pattern of how fast a cell grows with time is called 'mode of growth'. Mode of growth can be 'linear', when cells increase their size at a constant rate, or 'exponential', when cells increase their size at a rate proportional to their current size. A cell's mode of growth influences its inner workings, so identifying how a cell grows can reveal information about how a cell will behave.

Scientists can measure the size of cells as they age and identify their mode of growth using single cell imaging techniques. Unfortunately, the statistical methods available to analyze the large amounts of data generated in these experiments can lead to incorrect conclusions. Specifically, Kar et al. found that scientists had been using specific types of plots to analyze growth data that were prone to these errors, and may lead to misinterpreting exponential growth as linear and vice versa.

This discrepancy can be resolved by ensuring that the plots used to determine the mode of growth are adequate for this analysis. But how can the adequacy of a plot be tested? One way to do this is to generate synthetic data from a known model, which can have a specific and known mode of growth, and using this data to test the different plots. Kar et al. developed such a 'generative model' to produce synthetic data similar to the experimental data, and used these data to determine which plots are best suited to determine growth mode. Once they had validated the best statistical methods for studying mode of growth, Kar et al. applied these methods to growth data from the bacterium *Escherichia coli*. This showed that these cells have a form of growth called 'super-exponential growth'.

These findings identify a strategy to validate statistical methods used to analyze cell growth data. Furthermore, this strategy – the use of generative models to produce synthetic data to test the accuracy of statistical methods – could be used in other areas of biology to validate statistical approaches.

phenomenon of 'inspection bias' where certain bins have biased contributions. One of our main take home messages is the significance of having an underlying model (or models) to guide/test/validate data analysis methods. The underlying model is referred to as a generative model in the sense that it leads to similar data to that observed in the experiments. The importance of a so-called generative model has been beautifully advocated in the context of astrophysical data analysis (*Hogg et al., 2010*), yet biology brings in a plethora of exciting differences: while in physics noise from measurement instruments often dominates, in the biological examples we will dwell on here it is the *intrinsic* biological noise that can obscure the mathematical relation between variables when not handled properly. In the following, we will illustrate this rather philosophical introduction on a concrete and fundamental example, albeit e pluribus unum. We will focus on the analysis of the *Escherichia coli* growth curves obtained via high throughput optical microscopy. Nevertheless we anticipate the conceptual points made here – and demonstrated on a particular example of interest – will translate to other types of measurements, which make use of microscopy but also beyond.

Binning corresponds to grouping data based on the value of the x-axis variable, and finding the mean of the fluctuating y-axis variable for this group. By removing the fluctuations of the y-variable, the binning process often aims to expose the 'true' functional relation between the two variables which can be used to infer the underlying biological mechanism. While binning may provide a smooth non-linear relation between variables, linear regression is used to find a linear relationship between the variables. In addition to binning, we use the ordinary least squares regression where the slope and the intercept of the best linear fit line are obtained by minimizing the squared sum of the difference between the dependent variable raw data and the predicted value. Here, the best fit/the best linear fit is obtained using the raw data and not the binned data. Similar to binning, the assumption underlying linear regression is that our knowledge of x-axis variable is precise while the noise is in the y-axis variable.

It is important to discuss the sources of fluctuations in the y-axis variable before we proceed. In biology, fluctuations in the variables arise inevitably from the intrinsic variability within a cell population. Cells growing in the same medium and environment have different characteristics (e.g. growth rate) due to the stochastic nature of biochemical reactions in the cell (*Kiviet et al., 2014*). For example, the division event is controlled by stochastic reactions, whose variability leads to cell dividing at a size

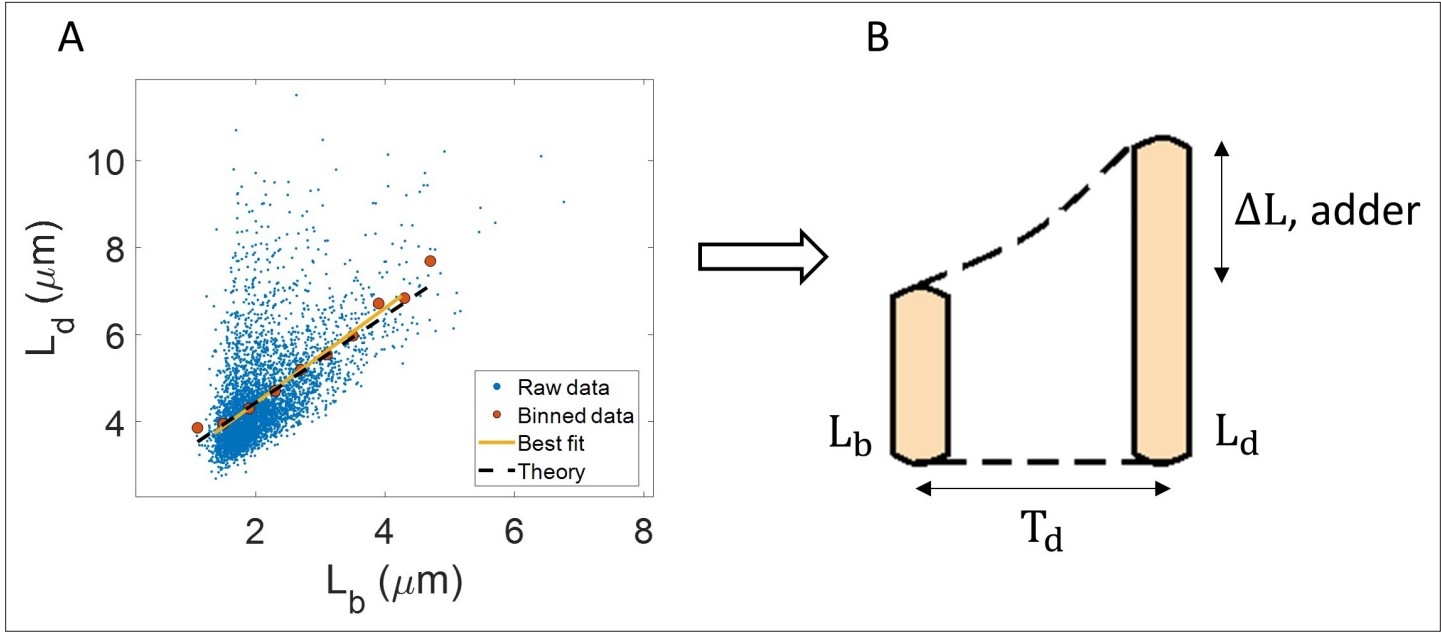

**Figure 1.** Utility of binning and linear regression. (**A**) Length at division ($L_d$) vs length at birth ($L_b$) is plotted using data obtained by *Tanouchi et al., 2017*. Raw data is shown as blue dots. We find the trend in binned data (red) to be linear with the underlying best linear fit (yellow) following the equation, $L_d = 1.09L_b + 2.24\mu m$. This is close to the adder behavior with an underlying equation given by $L_d = L_b + \Delta L$, where $\Delta L$ is the mean size added between birth and division (shown as black dashed line). B. A schematic of the adder mechanism is shown where the cell grows over its generation time ($T_d$) and divides after addition of length $\Delta L$ from birth. This ensures cell size homeostasis in single cells.

smaller or larger than the mean. In this paper, when modeling the data, we will consider the intrinsic noise as the only source of variability and assume that the measurement error is much smaller than the intrinsic variation in the population.

One example of the use of binning and linear regression is shown in *Figure 1A* where size at division ($L_d$) vs size at birth ($L_b$) is plotted using experimental data obtained by Tanouchi et al. for *E. coli* growing at 25 °C (*Tanouchi et al., 2017*). In *Figure 1A*, the functional relation between length at division and length at birth for *E. coli* is observed to be linear and close to $L_d = L_b + \Delta L$ (see the Experimental data section for details). The relation obtained allows us to hypothesize a coarse-grained biological model known as the adder model as shown in *Figure 1B* in which the length at division is set by addition of length $\Delta L$ from birth (*Soifer et al., 2016*; *Harris and Theriot, 2016*; *Si et al., 2019*; *Amir, 2014*; *Campos et al., 2014*; *Taheri-Araghi et al., 2015*; *Eun et al., 2018*). This previously discussed example demonstrates and reiterates the use of statistical analysis on single-cell data to understand the underlying cell regulation mechanisms. Using statistical methods such as binning and linear regression, other phenomenological models apart from adder have also been proposed in *E. coli* where the division length ($L_d$) is not directly 'set' by that at birth (*Ho and Amir, 2015*; *Micali et al., 2018*; *Witz et al., 2019*). The phenomenological models, in turn, can be related to mechanistic (molecular-level) models of cell size and cell cycle regulation (*Barber et al., 2017*). Recent work has shed light on the subtleties involved in interpreting the linear regression results for the $L_d$ vs $L_b$ plot where seemingly adder behavior in length can be obtained from a sizer model (division occurring on reaching a critical size) due to the interplay of multiple sources of variability (*Facchetti et al., 2019*). This issue is similar in spirit to those we highlight here.

The volume growth of single bacterial cells has been typically assumed to be exponential (*Godin et al., 2010*; *Wang et al., 2010*; *Campos et al., 2014*; *Cermak et al., 2016*; *Soifer et al., 2016*; *Iyer-Biswas et al., 2014*). Assuming ribosomes to be the limiting component in translation, growth is predicted to be exponential and growth rate depends on the active ribosome content in the cell (*Scott et al., 2010*; *Lin and Amir, 2018*; *Metzl-Raz et al., 2017*). Under the assumption of exponential growth, the size at birth ($L_b$), the size at division ($L_d$), and the generation time ($T_d$) are related to each other by,

$$\ln(\tfrac{L_d}{L_b}) = \lambda T_d, \qquad (1)$$

where $\lambda$ is the growth rate. Understanding the mode of growth is important for example, due to its potential effects on cell size homeostasis. Exponentially growing cells cannot employ a mechanism where they control division by timing a constant duration from birth but such a mechanism is possible in case of linear growth (*Amir, 2014*; *Kafri et al., 2016*; *Ho et al., 2018*). Linear regression performed on $\ln(\tfrac{L_d}{L_b})$ vs $\langle\lambda\rangle T_d$ plot, where $\langle\lambda\rangle$ is the mean growth rate, was used to infer the mode of growth in the archaeon *H. salinarum* (*Eun et al., 2018*), and in the bacteria *M. smegmatis* (*Logsdon et al., 2017*) and *C. glutamicum* (*Messelink et al., 2020*), for example. If the best linear fit follows the y = x trend, the resulting functional relation might point to growth being exponential. A corollary to this is the rejection of exponential growth when the slope and intercept of the best linear fit deviate from one and zero, respectively (*Messelink et al., 2020*). Thus, binning and linear regression applied on single-cell data appear to provide information about the underlying biology, in this case, the mode of cellular growth. We will test the validity of such inference by analyzing synthetic data generated using generative models. We find that linear regression performed on the plot $\ln(\tfrac{L_d}{L_b})$ vs $\langle\lambda\rangle T_d$, surprisingly, does not provide information about the mode of growth. Nonetheless, we show that other methods of statistical analysis such as binning growth rate vs age plots are adequate in addressing the problem. Using these validated methods on experimental data, we find that *E. coli* grows non-exponentially. In later stages of the cell cycle, the growth rate is higher than that in early stages.

## Results
### Statistical methods like binning and linear regression should be interpreted based on a model

To illustrate the pitfalls associated with binning, we use data from recent experiments on *E. coli* where the length at birth, the length at division and the generation time were obtained for multiple cells (see Experimental methods and [*Tiruvadi-Krishnan et al., 2021*]). Phase-contrast microscopy was used to obtain cell length at equal intervals of time. Note that we consider length to reflect cell size in this paper rather than other cell geometry characteristics such as surface area and volume. The length growth rate that we elucidate in the paper can be different from the cell volume growth rate as shown in Appendix 1 assuming a simple cell morphology and exponential growth. Using the same cell morphology, we also find the length growth rate to be identical to cell surface growth rate. To investigate if the cell growth was exponential, we plotted $\ln(\tfrac{L_d}{L_b})$ vs $\langle\lambda\rangle T_d$ for cells growing in M9 alanine minimal medium at 28 °C ($\langle T_d\rangle$ = 214 min). The linear regression of these data yields a slope of 0.3 and an intercept of 0.4 as shown in *Figure 2A*. The binned data and the best linear fit deviate significantly from the y = x line (see *Supplementary file 1*). Additionally, the binned data follows a non-linear trend and flattens out at longer generation times. We also found similar deviations in the binned data and best linear fit in glycerol medium ($\langle T_d\rangle$ = 164 min) shown in *Figure 2—figure supplement 1A*, and glucose-cas medium ($\langle T_d\rangle$ = 65 min) shown in *Figure 2—figure supplement 1B*. Qualitatively similar results have been recently obtained for another bacterium, *C. glutamicum*, in *Messelink et al., 2020*. These results might point to growth being non-exponential.

Next, we will approach the same problem but with a generative model. We will first show that the $\ln(\tfrac{L_d}{L_b})$ vs $\langle\lambda\rangle T_d$ binned plot could not distinguish exponential growth from non-exponential growth. For that purpose, we use a previously studied model (*Eun et al., 2018*) which considers growth to be exponential with the growth rate distributed normally and independently between cell cycles with mean growth rate $\langle\lambda\rangle$ and standard deviation $CV_\lambda\langle\lambda\rangle$. $CV_\lambda$ is thus the coefficient of variation (CV) of the growth rate and is assumed to be small. To maintain a narrow distribution of cell size, cells must employ regulatory mechanisms. In our model, we assume that, barring the noise due to stochastic biochemical reactions, cells attempt to divide at a particular size $L_d$ given size at birth $L_b$. Keeping the model as generic as possible, we can write $L_d$ as a function of $L_b$, f($L_b$) which can be thought of as a coarse-grained model for the regulatory mechanism. *Amir, 2014* provides a framework to capture the regulatory mechanisms by choosing f($L_b$) = 2 $L_b^{1-\alpha}L_0^\alpha$. $L_0$ is the typical size at birth and $\alpha$, which can take values between 0 and 2, reflects the strength of regulation strategy. $\alpha = 0$ corresponds to the timer model where division occurs on average after a constant time from birth, and $\alpha = 1$ is the sizer model where a cell divides upon reaching a critical size. $\alpha = 1/2$ can be shown to be equivalent to the

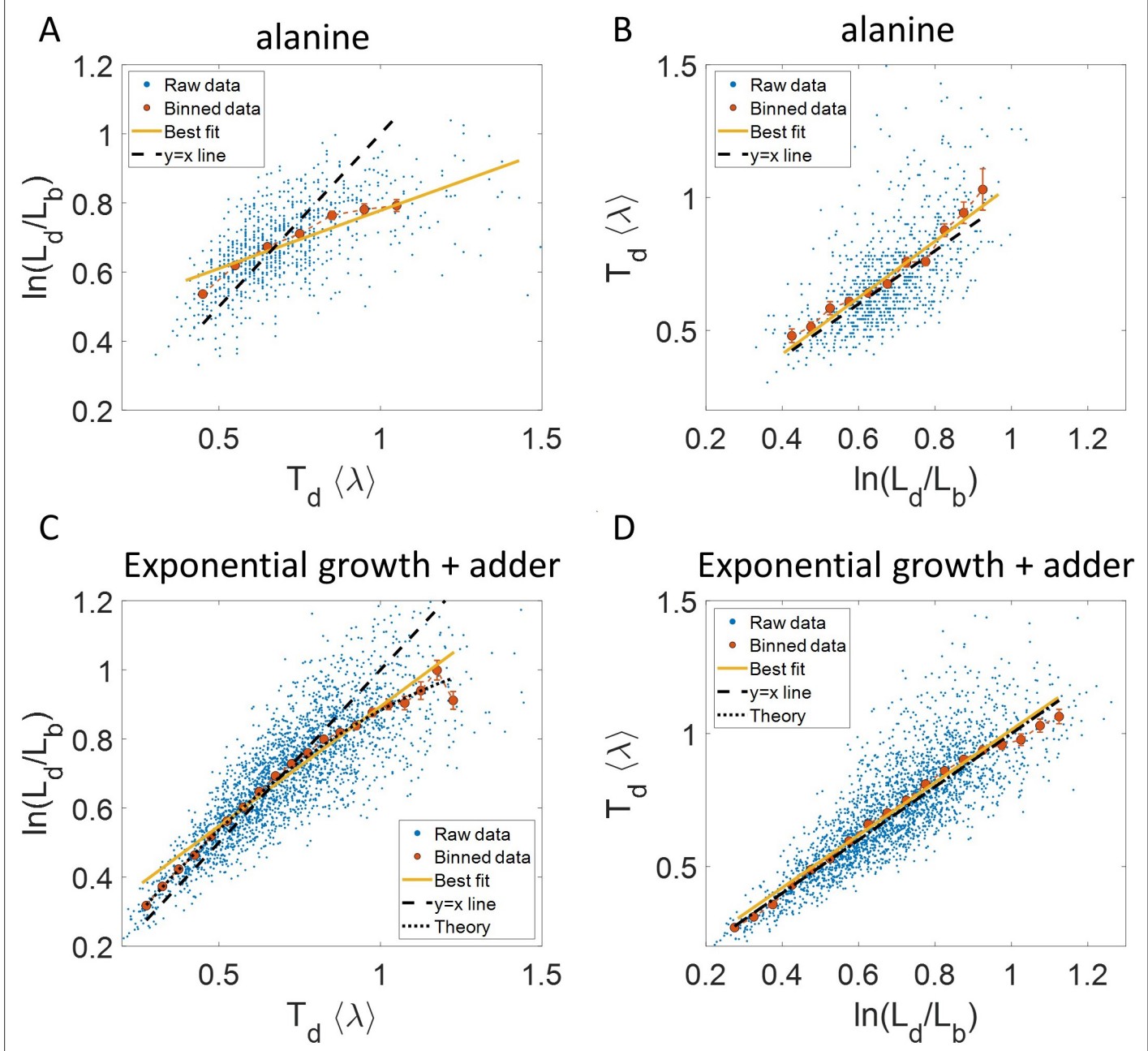

**Figure 2.** Plots that could potentially lead to misinterpreting exponential growth. (**A, B**) Data is obtained from experiments in M9 alanine medium ($\langle T_d \rangle$ = 214 min, N = 816 cells). (**A**) $\ln(\frac{L_d}{L_b})$ vs $\langle \lambda \rangle T_d$ plot is shown. The blue dots are the raw data, the red correspond to the binned data trend, the yellow line is the best linear fit obtained by performing linear regression on the raw data and the black dashed line is the y = x line. A priori, non-linear trend in binned data might point to growth being non-exponential. (**B**) $\langle \lambda \rangle T_d$ vs $\ln(\frac{L_d}{L_b})$ plot is shown for the same experiments. (**C, D**) Simulations of exponentially growing cells following the adder model are carried out for N = 2500 cells. The parameters used are provided in the Simulations section. (**C**) $\ln(\frac{L_d}{L_b})$ vs $\langle \lambda \rangle T_d$ plot is shown. The trend in binned data shown in red is non-linear and the best linear fit of raw data (yellow) deviates from the y = x line (black dashed line). The black dotted line is the expected trend obtained from theory (**Equation 2**). For parameters used in the simulations here, the black dotted line follows $\ln(\frac{L_d}{L_b}) = 1.26\langle \lambda \rangle T_d - 0.38(\langle \lambda \rangle T_d)^2$. (**D**) $\langle \lambda \rangle T_d$ vs $\ln(\frac{L_d}{L_b})$ plot is shown with binned data in red and the best linear fit on raw data in yellow closely following the expected trend of y = x line (black dashed line). The theoretical binned data trend (black dotted line) is expected to follow the y = x trend. In all of these plots, the binned data is shown only for those bins with more than 15 data points in them.

The online version of this article includes the following figure supplement(s) for figure 2:

**Figure supplement 1.** Experimental data: $\ln(\frac{L_d}{L_b})$ vs $\langle \lambda \rangle T_d$ (left) and $\langle \lambda \rangle T_d$ vs $\ln(\frac{L_d}{L_b})$ plot (right) is shown for, (**A**).

**Figure supplement 2.** Binned data trend in growth rate ($\lambda$) and inverse generation time ($\frac{1}{T_d}$) plots.

adder model where division is controlled by addition of constant size from birth (**Amir, 2014**). In addition to the deterministic function (f) specifying division, the size at division is affected by noise ($\frac{\zeta}{\langle\lambda\rangle}$) in division timing. We assume it has a Gaussian distribution with mean zero and standard deviation $\frac{\sigma_n}{\langle\lambda\rangle}$ and that it is independent of the growth rate. Thus, the generation time ($T_d$) can be mathematically written as $T_d = \frac{1}{\lambda}\ln(\frac{f(L_b)}{L_b}) + \frac{\zeta}{\langle\lambda\rangle}$ and is influenced by growth rate noise and division timing noise. Note that replacing the time additive division timing noise with a size additive division timing noise will not affect the results qualitatively (see 'Model' and 'Exponential growth' sections for details and **Supplementary file 1** for variable definitions).

For perfectly symmetrically dividing cells whose sizes are narrowly distributed, we find the trend in the binned data for $\ln(\frac{L_d}{L_b})$ vs $\langle\lambda\rangle T_d$ plot to be (see section 'Predicting the results of statistical constructs applied on $\ln(\frac{L_d}{L_b})$ vs $\langle\lambda\rangle T_d$ and $\langle\lambda\rangle T_d$ vs $\ln(\frac{L_d}{L_b})$'),

$$y = x\left(1 + \frac{1 - \frac{x}{\ln(2)}}{1 + \frac{2}{2-\alpha}\frac{\sigma_n^2}{CV_\lambda^2\ln^2(2)}}\right).\tag{2}$$

Fixing $CV_\lambda = \sigma_n = 0.15$, we show using simulations in **Figure 2C** the non-linear trend in the binned data even though we assumed exponential growth. Similarly, on performing linear regression on the raw data of $\ln(\frac{L_d}{L_b})$ vs $\langle\lambda\rangle T_d$ plot, we find that the slope of the best linear fit is not equal to one and the intercept is non-zero (see **Equation 27** and **28** and **Figure 2C**). **Equation 2** shows that the trend in the binned data depends on the ratio of growth rate noise and division timing noise. The slope is equal to one and intercept is zero only if the noise in growth rate is negligible as compared to the division timing noise. In experiments that is rarely the case, hence, the binned data trend and the best linear fit deviate from the y = x line even though growth might be exponential. Thus, we cannot rule out exponential growth in the *E. coli* experiments despite the binned data trend being non-linear and the best-fit line deviating from the y = x line.

Why does a non-linear relationship in the binned data for the plot $\ln(\frac{L_d}{L_b})$ vs $\langle\lambda\rangle T_d$ arise even for exponential growth? According to the model, $L_d$ is determined by a deterministic strategy, f($L_b$) and a time/size additive division timing noise. The noise component which affects $L_d$ and subsequently the quantity $\ln(\frac{L_d}{L_b})$ is thus the noise in division timing and not the growth rate. The generation time ($T_d$) plotted on the x-axis is influenced by the noise in division timing as well as the noise in growth rate. Binning assumes that for a fixed value of the x-axis variable, the noise from other sources affects only the y-axis variable (the binned variable). Similarly for linear regression, the underlying assumption is that the independent variable on x-axis is precisely known while the dependent variable on the y-axis is influenced by the independent variable and from external factors other than the independent variable. In this case, only $\langle\lambda\rangle T_d$ plotted on x-axis is influenced by growth rate noise while both $\langle\lambda\rangle T_d$ and $\ln(\frac{L_d}{L_b})$ are influenced by noise in division time. This does not fit the assumption for binning and linear regression and hence, the best linear fit for $\ln(\frac{L_d}{L_b})$ vs $\langle\lambda\rangle T_d$ plot might deviate from the y = x line even in the case of exponential growth.

Another way of explaining the deviation from the linear y = x trend is by inspection bias, which arises when certain data is over-represented (**Stein and Dattero, 2018**). Cells which have a longer generation time than the mean will most likely have a slower growth rate. Thus, in **Figure 2A and C**, at larger values of $\langle\lambda\rangle T_d$ or $T_d$, the bin averages are biased by slower growing cells, thus making $\ln(\frac{L_d}{L_b})$ or $\lambda T_d$ to be lower than expected. This provides an explanation for the flattening of the trend.

It follows from the previous discussion that if one bins data by $\ln(\frac{L_d}{L_b})$ then the assumption for binning is met. Both of the variables $\langle\lambda\rangle T_d$ and $\ln(\frac{L_d}{L_b})$ are influenced by the noise in division time but $\langle\lambda\rangle T_d$ plotted on the y-axis is also influenced by the growth rate noise. Thus, the y-axis variable, $\langle\lambda\rangle T_d$ is determined by the x-axis variable, $\ln(\frac{L_d}{L_b})$, and an external source of noise, in this case, the growth rate noise. Thus, based on our model, we expect the trend in binned data and linear regression performed on the interchanged axes to follow the y = x trend for exponentially growing cells (see section 'Predicting the results of statistical constructs applied on $\ln(\frac{L_d}{L_b})$ vs $\langle\lambda\rangle T_d$ and $\langle\lambda\rangle T_d$ vs $\ln(\frac{L_d}{L_b})$'). Indeed, on interchanging the axis and plotting $\langle\lambda\rangle T_d$ vs $\ln(\frac{L_d}{L_b})$ for synthetic data, we find that the trend in the binned data and the best linear fit closely follows the y = x line (**Figure 2D**). We also find that the best linear fit follows the y = x line in the case of alanine (**Figure 2B**), glycerol (**Figure 2—figure supplement 1A**) and glucose-cas (**Figure 2—figure supplement 1B**). A change from non-linear

behavior to that of linear on interchanging the axes is also observed in a related problem where growth rate ($\lambda$) and inverse generation time ($\frac{1}{T_d}$) are considered (*Figure 2—figure supplement 2* and Section 'Interchanging axes in growth rate vs inverse generation time plot might lead to different interpretations').

Thus far, we showed for a range of models where birth controls division that the binned data trend for $\ln(\frac{L_d}{L_b})$ as function of $\langle\lambda\rangle T_d$ is non-linear and dependent on the noise ratio $\frac{\sigma_n}{CV_\lambda}$ in the case of exponential growth. On interchanging the axes the binned data trend agrees with the y = x line independent of the growth rate and division time noise. However, we will show next that this agreement with the y = x trend cannot be used as a 'smoking gun' for inferring exponential growth from the data.

To investigate this further, let us consider linear growth, which has also been suggested to be followed by *E. coli* cells (*Mitchison, 2005*; *Abner et al., 2014*). The underlying equation for linear growth is,

$$L_d - L_b = \lambda' T_d, \tag{3}$$

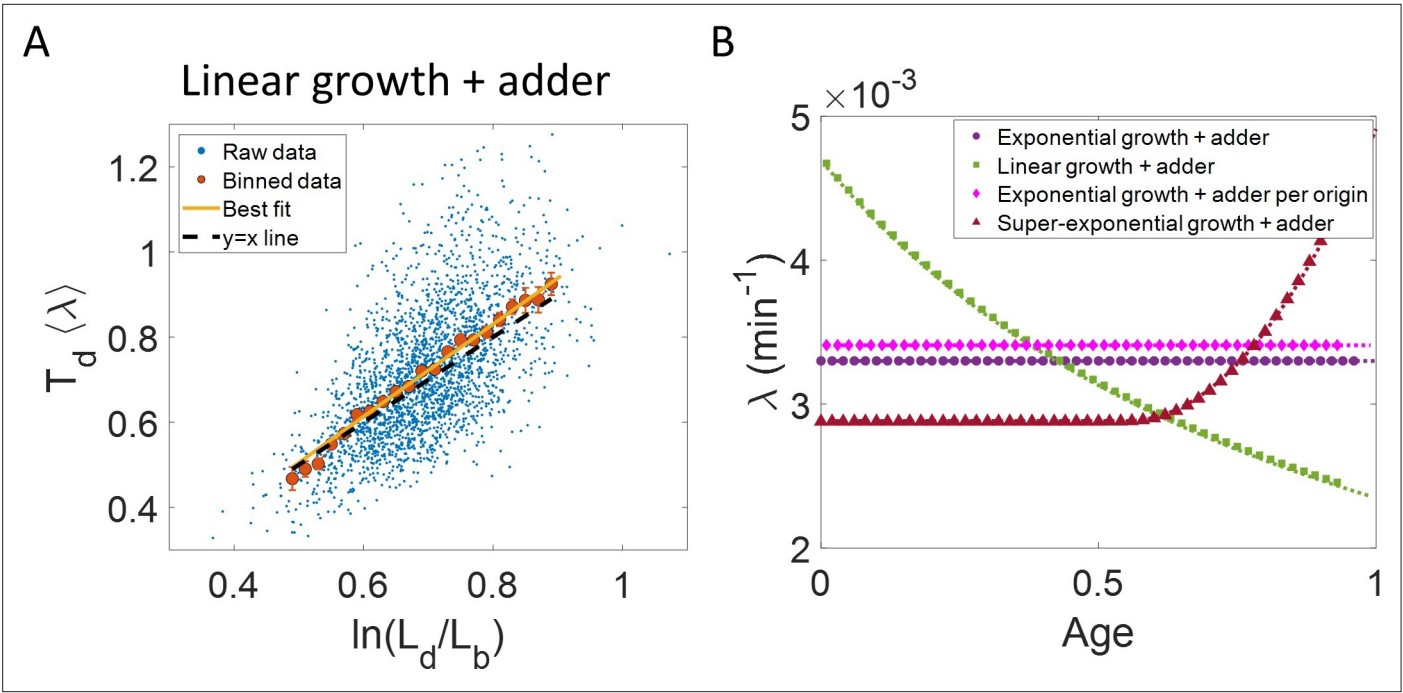

**Figure 3.** Differentiating linear growth from exponential growth. (**A**) $\langle\lambda\rangle T_d$ vs $\ln(\frac{L_d}{L_b})$ plot is shown for simulations of linearly growing cells following the adder model for N = 2500 cell cycles. The binned data (red) and the best linear fit on raw data (yellow) closely follows the y = x trend (black dashed line) which could be incorrectly interpreted as cells undergoing exponential growth. (**B**) The binned data trend for growth rate vs age plot is shown as purple circles for simulations of N = 2500 cell cycles of exponentially growing cells following the adder model. We observe the trend to be nearly constant as expected for exponential growth (purple dotted line). Since the growth rate is fixed at the beginning of each cell cycle in the above simulations, we do not show error bars for each bin within the cell cycle. Also shown as green squares is the growth rate vs age plot for simulations of N = 2500 cell cycles of linearly growing cells following the adder model. As expected for linear growth, the binned growth rate decreases with age as $\lambda \propto \frac{1}{1+age}$ (green dotted line). The binned growth rate trend (shown as magenta diamonds) is also found to be nearly constant as expected (shown as magenta dotted line) for the simulations of exponentially growing cells following the adder per origin model. We also show that the binned growth rate trend (red triangles) increases for simulations of the adder model with the cells undergoing faster than exponential growth. The trend is in agreement with the underlying growth rate function (shown as red dotted line) used in the simulations of super-exponential growth. Thus, the plot growth rate vs age provides a consistent method to identify the mode of growth. Parameters used in the above simulations of exponential, linear and super-exponential growth are derived from the experimental data in alanine medium. Details are provided in the Simulations section.

The online version of this article includes the following figure supplement(s) for figure 3:

**Figure supplement 1.** Predicting statistics based on a model of linear growth.

**Figure supplement 2.** Inspection bias in the growth rate vs time plots obtained from simulations.

**Figure supplement 3.** Differential methods of quantifying growth.

where $\lambda'$ is the the elongation speed that is, $\frac{dL}{dt}$. For cells growing linearly, the best linear fit for the plot $\langle\lambda\rangle T_d$ vs $\ln(\frac{L_d}{L_b})$ is expected to deviate from the y = x line. As before, we fix $\langle\lambda\rangle$ to be the mean of $\frac{1}{T_d}\ln(\frac{L_d}{L_b})$, agnostic of the linear mode of growth. Surprisingly, we found that for the class of models where birth controls division by a strategy f($L_b$) and cells grow linearly, the best linear fit for $\langle\lambda\rangle T_d$ vs $\ln(\frac{L_d}{L_b})$ agrees closely with the y = x trend. On carrying out analytical calculations based on this model, we obtain the slope and the intercept of the $\langle\lambda\rangle T_d$ vs $\ln(\frac{L_d}{L_b})$ plot to be $\frac{3}{2}\ln(2) \approx 1.04$ and –0.03 respectively, which is very close to that for exponential growth (see section 'Differentiating linear from exponential growth'). This is shown for simulations of linear growth with cells following an adder model in *Figure 3A*. Given no information about the underlying model, *Figure 3A* could be interpreted as cells undergoing exponential growth contrary to the assumption of linear growth in simulations. Thus, when handling experimental data, cells undergoing either exponential or linear growth might seem to agree closely with the y = x trend. *Deforet et al., 2015* used the linear binned data trend in case of $\langle\lambda\rangle T_d$ vs $\ln(\frac{L_d}{L_b})$ plot to infer exponential growth but as we showed in this section, the linear trend does not rule out linear growth. This again reiterates our message of having a generative model to guide the data analysis methods such as binning and linear regression. For completeness, we also test the utility of $\ln(\frac{L_d}{L_b})$ vs $\langle T_d\rangle\lambda$ and its interchanged axes plots to elucidate the mode of growth (Appendix 2). We find that binning and linear regression applied on these plots can not differentiate between exponential and linear growth.

To conclude the discussion of linear growth, we note that the natural plot for this growth regime is $\langle\lambda_{lin}\rangle T_d$ vs $l_d - l_b$ and the plot obtained on interchanging the axes (see the Linear growth section and *Figure 3—figure supplement 1A and B*). Here $l_b$, $l_d$ and $\lambda_{lin}$ are defined to be quantities $L_b$, $L_d$ and $\lambda'$, respectively, normalized by the mean length at birth. For cells growing exponentially, the best linear fit for the $\langle\lambda_{lin}\rangle T_d$ vs $l_d - l_b$ plot is expected to deviate from the y = x line. This is indeed what is observed in *Figure 3—figure supplement 1* where simulations of exponentially growing cells following the adder model are presented (see 'Differentiating linear from exponential growth' for extended discussion).

In all the cases above, the problem at hand deals with distilling the biologically relevant functional relation between two variables. However, the data is assumed to be subjected to fluctuations of various sources, and it is important to ensure that the statistical construct we are using (e.g. binning) is robust to these. How can we know a priori whether the statistical method is appropriate and a 'smoking gun' for the functional relation we are conjecturing? The examples shown above suggest that performing statistical tests on synthetic data obtained using a generative model is a convenient and powerful approach. Note that in cases such as the ones studied here where analytical calculations may be performed, one may not even need to perform any numerical simulations to test the validity of the methods.

## Growth rate vs age plots are consistent with the underlying growth mode

In the last section, we showed that the plots $\ln(\frac{L_d}{L_b})$ vs $\langle\lambda\rangle T_d$ and $\langle\lambda\rangle T_d$ vs $\ln(\frac{L_d}{L_b})$ are not decisive in identifying the mode of growth. Recent works on *B. subtilis* (*Nordholt et al., 2020*) and fission yeast (*Knapp et al., 2019*) have used differential methods of quantifying growth namely growth rate ( = $\frac{1}{L}\frac{dL}{dt}$) vs age plots and elongation speed (=$\frac{dL}{dt}$) vs age plots to probe the mode of growth within a cell cycle. Here, $L$ denotes the size of the cell after time $t$ from birth in the cell cycle and age denotes the ratio of time $t$ to $T_d$ within a cell cycle (hence it ranges from 0 to 1 by construction within a cell cycle). In this section, using various models of cell growth and cell cycle, we test the growth rate vs age method. Note that the growth rate vs age and the elongation speed vs age plots are not dimensionless unlike the previous plots. Using the growth rate vs age and elongation speed vs age plots, we aim to quantify the growth rate changes within a cell cycle. For cells assumed to be growing exponentially, growth rate is constant throughout the cell cycle. On averaging over multiple cell cycles, the trend of binned data is expected to be a horizontal line with value equal to mean growth rate which is indeed what we find in the numerical simulations of the adder and the adder per origin model (*Ho and Amir, 2015*), as shown in *Figure 3B*. The binned data trend in each of the models matches the theoretical predictions of growth rate (shown as dotted lines). In contrast, for linearly growing cells, the elongation speed is expected to remain constant. We show this constancy using numerical simulations of linearly growing cells following the adder model (*Figure 3—figure supplement 3A*). In accordance with this result, the

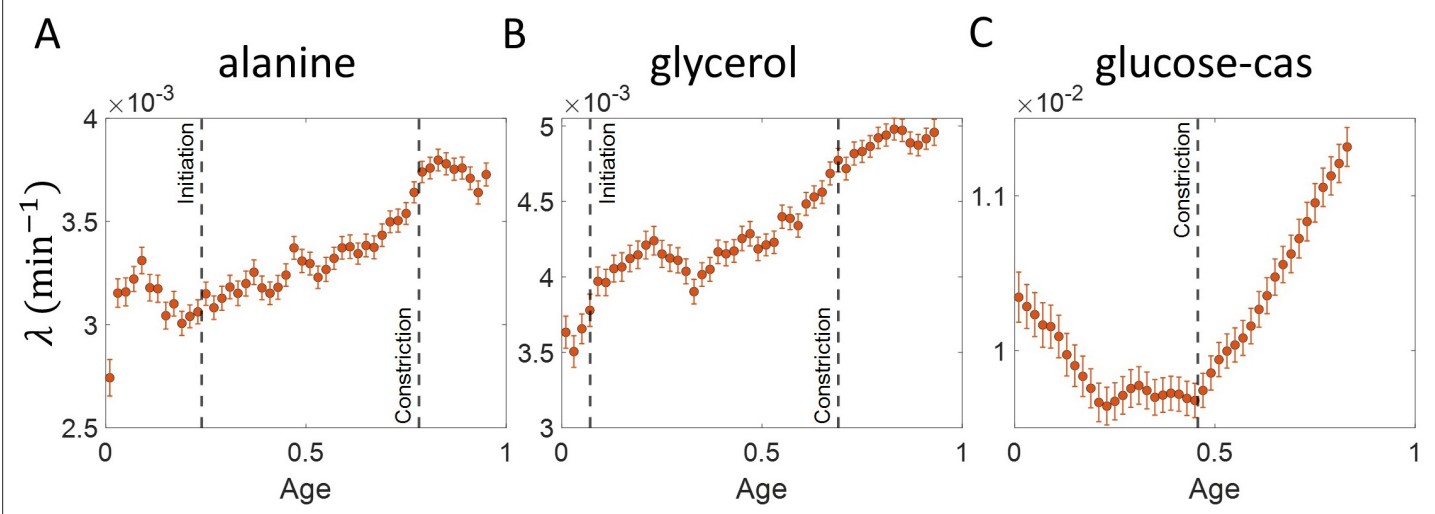

**Figure 4.** Growth rate vs age obtained from experiments: Growth rate vs age plots are shown for *E. coli* experimental data. The red dots correspond to the binned data trends showing the variation in growth rate. The medium in which the experiments were conducted are (**A**) Alanine ($\langle T_d \rangle$ = 214 min) (**B**) Glycerol ($\langle T_d \rangle$ = 164 min) (**C**) Glucose-cas ($\langle T_d \rangle$ = 65 min). The error bars show the standard deviation of the growth rate in each bin scaled by $\frac{1}{\sqrt{N}}$, where N is the number of cells in that bin. The dashed vertical lines mark the age at initiation of DNA replication (left line) and the start of septum formation (right line). In case of glucose-cas, the initiation age is not marked as it occurs in the mother cell.

The online version of this article includes the following figure supplement(s) for figure 4:

**Figure supplement 1.** Growth rate vs age curves extended beyond the division event.

**Figure supplement 2.** Inspection bias in the growth rate vs time from constriction plots obtained from experiments.

growth rate is expected to decrease with cell age as $\lambda \propto \frac{1}{1+age}$. This is verified in ***Figure 3B*** by again using the numerical simulations of linear growth with cells following the adder model. The binned data trend for linear growth (green squares) matches the theoretical predictions of $\lambda \propto \frac{1}{1+age}$ (green dotted line).

Thus, the two growth modes (exponential and linear) could be differentiated using the growth rate vs age plot (for details see 'Growth rate vs age and elongation speed vs age plots' section). However, the growth rate vs age plots can be used to infer the mode of growth beyond the two discussed above. We show this by using simulations of cells following the adder model and undergoing faster than exponential or super-exponential growth (see the Simulations section for details). In such a case, the growth rate is expected to increase. This increase in growth rate is shown in ***Figure 3B*** using simulations. The binned data trend (red triangles) again matches the growth rate mode used in the simulations (red dotted line). Thus, the growth rate vs age plots are a consistent method to distinguish linear from exponential and super-exponential growths.

Using the validated growth rate vs age plots, we obtained the growth rate trend for experimental data on *E. coli* for the three growth conditions studied in this paper (***Figure 4A-C***). We found an increase in growth rate in all growth conditions during the course of the cell cycle. One may wonder whether such an increase may be explained by the *E. coli* morphology alone, due to the presence of hemispherical poles. For exponentially growing cell volume and considering a geometry of *E. coli* with spherical caps at the poles, the percentage increase in the growth rate of length over a cell cycle is around 3 % which is significantly smaller than that observed in our experimental data. Considering cell size trajectories (cell size, *L* at time, *t* data) where cell lengths were tracked beyond the cell division event (by considering cell size in both daughter cells), we also found that the growth rate decreases close to division (age ≈ 1) and returns to a value nearly equal to that observed at the beginning of cell cycle (age ≈ 0) as shown in ***Figure 4—figure supplement 1*** (see 'Growth rate vs age and elongation speed vs age plots' section for extended discussion).

The above question of mode of growth within a cell cycle can also be analyzed in relation to a specific event. Several studies have pointed to a change in growth rate at the onset of constriction (***Reshes et al., 2008***; ***Banerjee et al., 2017***). This change in growth rate can be probed using growth

rate vs time plots where time is taken relative to the onset of constriction as shown in *Figure 4—figure supplement 2*. These plots show a decrease in growth rates at the two extremes of the plot. These decreases are due to inspection bias, where the growth rate trend is affected by the biased contribution of cells with a higher than average generation time or equivalently slower growth rate (see 'Growth rate vs time from specific event plots are affected by inspection bias' section for extended discussion). Inspection bias is also observed when timing is considered relative to other cell events such as cell birth (see 'Growth rate vs time from specific event plots are affected by inspection bias' section and *Figure 3—figure supplement 2*).

It might not always be possible to obtain growth rate trajectories as a function of time/cell age. Godin et al. instead obtained the instantaneous biomass growth speed ($\frac{dM}{dt}$) as a function of its buoyant mass ($M$) (*Godin et al., 2010*). On applying linear regression for instantaneous mass growth speed vs mass, we expect the slope of the best linear fit obtained to provide the average growth rate ($\langle\langle\frac{1}{M}\frac{dM}{dt}\rangle\rangle$) under the assumption of exponential growth while for linear growth the intercept provides the average growth speed. Using this method, biomass was suggested to be growing exponentially. This method can be applied to study the length growth rate within the cell cycle by plotting elongation speed as a function of length (*Cadart et al., 2019*). We find that the binned data trend and the best linear fit of this plot follow the expected trend for linear and exponential growth as shown in *Figure 3—figure supplement 3B* and *Figure 3—figure supplement 3D*, respectively, for a cell cycle model where division is controlled via an adder mechanism from birth. However, the trend obtained appears to be model-dependent as shown in *Figure 3—figure supplement 3F* where the underlying cell cycle model used in the simulations is the adder per origin model. For this model, the binned data trend is found to be non-linear with the growth rate speeding up at large sizes, despite the synthetic data being generated for perfectly exponential growth. This non-linear trend can lead to growth rate being misinterpreted as non-exponential within the cell cycle (see 'Results of elongation speed vs size plots are model-dependent' section for details). Thus, an analysis using the elongation speed vs size plot must be accompanied with an underlying cell cycle model.

In summary, we found that the growth rate vs age plot was a consistent method to determine the changes in growth rate within a cell cycle. Unlike the growth rate vs age plots, the inference from the growth rate vs size plots was found to be model-dependent. Using the growth rate vs age plots, we show that the length growth of *E. coli* can be faster than exponential.

## Discussion

Statistical methods such as binning and linear regression are useful for interpreting data and generating hypotheses for biological models. However, we show in this paper that predicting the relationships between experimentally measured quantities based on these methods might lead to misinterpretations. Constructing a generic model and verifying the statistical analysis on the synthetic data generated by this model provides a more rigorous way to mitigate these risks.

In the paper, we provide examples in which $\ln(\frac{L_d}{L_b})$ vs $\langle\lambda\rangle T_d$ and $\langle\lambda\rangle T_d$ vs $\ln(\frac{L_d}{L_b})$ plots fail as a method to infer the mode of growth. The binned data trend and the best linear fit for the $\ln(\frac{L_d}{L_b})$ vs $\langle\lambda\rangle T_d$ plot was found to be dependent upon the noise parameters in the class of models where birth controlled division (*Equation 2*). We also show that $\langle\lambda\rangle T_d$ vs $\ln(\frac{L_d}{L_b})$ plot could not differentiate between exponential and linear modes of growth (*Figures 2D and 3A*). Thus, we conclude that the best linear fit for the above plots might not be a suitable method to infer the mode of growth but they are just one of the many correlations which the correct cell cycle model should be able to predict.

We found growth rate vs age and elongation speed vs age plots to be consistent methods to probe growth within a cell cycle. The method was validated using simulations of various cell cycle models (such as the adder, and adder per origin model, where in the latter, control over division is coupled to DNA replication) and the binned growth rate trend agreed closely with the underlying mode of growth for the wide range of models considered (*Figure 3B*). In the case of growth rate vs time plots, it was important to take into consideration the effects of inspection bias. We used cell cycle models to show the time regimes where inspection bias could be observed (*Figure 3—figure supplement 2*). In the regime with negligible inspection bias, we could reconcile the growth rate trend obtained using growth rate vs age (*Figure 4A-C*) and growth rate vs time plots (*Figure 4—figure supplement 2*). The authors in *Messelink et al., 2020* circumvent inspection bias in the elongation speed vs time from birth plots by focusing their analysis on the time period from cell birth to the generation time of the

fastest dividing cell. The authors of *Panlilio et al., 2021*, while investigating the division behavior in the cells undergoing nutrient shift within their cell cycle, use both models and experimental data from steady-state conditions to identify inspection bias. These serve as good examples of using models to aid data analysis.

Statistics obtained from linear regression such as in *Figure 1A* help narrow down the landscape of cell cycle models, but many have potential pitfalls lurking which might lead to misinterpretations (*Figures 2C and 3A*). There are additional issues beyond those concerning linear regression and binning discussed here. For example, *Willis and Huang, 2017* discusses Simpson's paradox (*Simpson, 1951*) where distinct cellular sub-populations might lead to erroneous interpretation of cell cycle mechanisms. Examples of such distinct sub-populations are found in asymmetrically dividing bacteria such as *M. smegmatis* (*Aldridge et al., 2012*; *Logsdon et al., 2017*). Another source of misinterpretation could arise from presence of measurement errors. Throughout this work, we deal with intrinsic noise and neglect measurement error. However, when measurement noise affects both x-axis and y-axis variables, the slope of the best linear fit is biased towards zero. This can lead to potentially related variables being misinterpreted as uncorrelated. Measurement errors can, however, be handled based on a model. Using a model which includes measurement error as a source of noise, we can guide the binning analysis. Using this methodology, we verified that typical measurement errors ($\approx 0.02 L_b$) *Messelink et al., 2020*; *Kaiser et al., 2018* have negligible effects on the growth rate trends obtained from the experimental data used in our work.

Single cell size in *E. coli* has been reported to grow exponentially (*Campos et al., 2014*; *Wang et al., 2010*; *Cermak et al., 2016*; *Soifer et al., 2016*; *Iyer-Biswas et al., 2014*; *Godin et al., 2010*), linearly (*Mitchison, 2005*), bilinearly (*Kubitschek, 1981*) or trilinearly (*Reshes et al., 2008*). These are inconsistent with our observations in *Figure 4A-C* where we find that growth can be super-exponential. The non-monotonic behavior in the fastest-growth condition is reminiscent of the results reported in *Nordholt et al., 2020* for *B. subtilis*. The authors of *Nordholt et al., 2020* attribute the increase in growth rate to a multitude of cell cycle processes such as initiation of DNA replication, divisome assembly, septum formation. In the two slower growth conditions (*Figure 4A-B*), we find that the growth rate increase starts before the time when the septal cell wall synthesis starts i.e., the constriction event. However, in the fastest growth condition (*Figure 4C*), the timing of growth rate

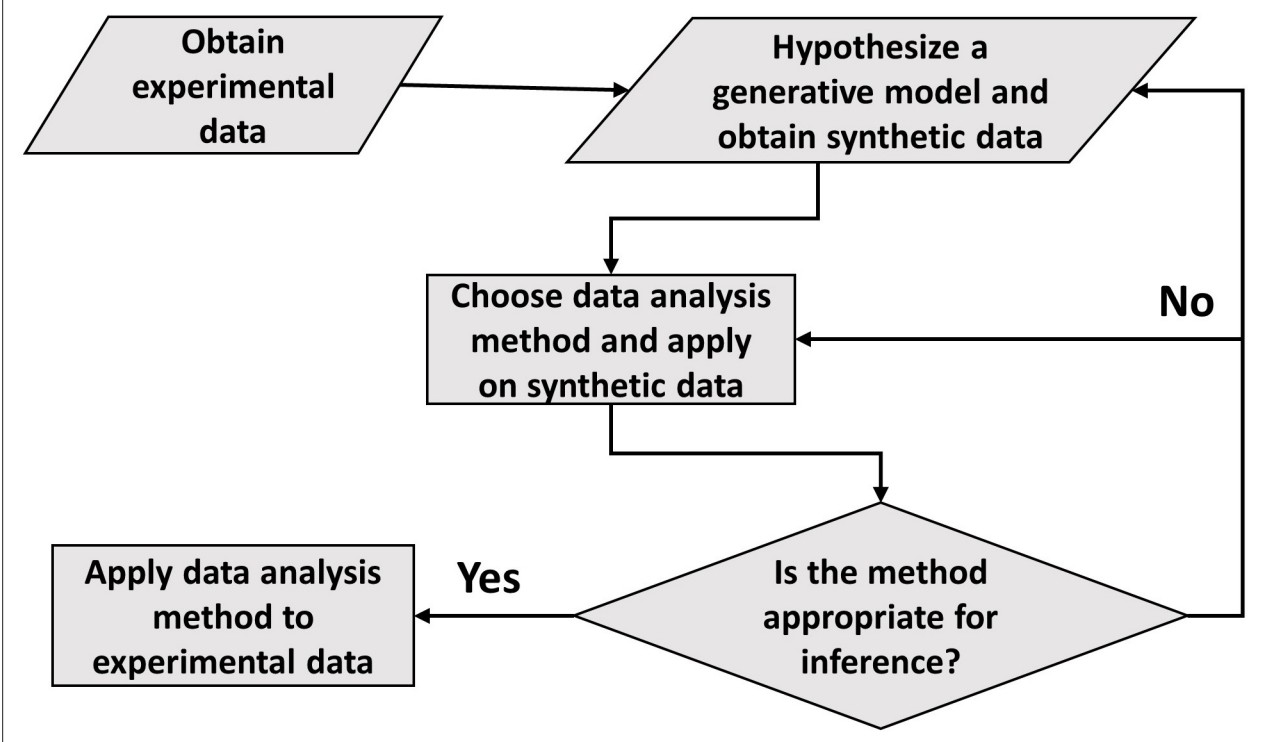

**Figure 5.** A flowchart of the general framework proposed in the paper to carry out data analysis.

increase seems to coincide with the onset of constriction which is in agreement with previous findings (*Reshes et al., 2008*; *Banerjee et al., 2017*).

It is important to distinguish between length growth and biomass growth. *Oldewurtel et al., 2021* measures biomass and cell volume and finds the mass-density variations within the cell-cycle to be small. In this paper, since we observe the length growth to be non-exponential (*Figure 4*), it remains to be seen whether biomass growth also follows a similar non-exponential behavior or if it is exponential as previously suggested (*Godin et al., 2010*; *Oldewurtel et al., 2021*).

In conclusion, the paper draws the attention of the readers to the careful use of statistical methods such as linear regression and binning. Although shown in relation to cell growth, this approach to data analysis seems ubiquitous. The general framework of carrying out data analysis is presented in *Figure 5*. It proposes the construction of a generative model based on the experimental data collected. Of course, we do not always know whether the model used is an adequate description of the system. What is the fate of the methodology described here in such cases? First, we should be reminded of Box's famous quote 'all models are wrong, some are useful'. The goal of a model is not to provide as accurate a description of a system as possible, but rather to capture the essence of the phenomena we are interested in and stimulate further ideas and understanding. In our context, the goal of the model is to provide a rigorous framework in which data analysis tools can be critically tested. If verified within the model, it is by no means proof of the success of the model and the method itself, and further comparisons with the data may falsify it leading to the usual (and productive) cycle of model rejection and improvement via comparison with experiments. However, if the best model we have at hand shows that the data analysis method is non-informative, as we have shown here on several methods used to identify the mode of growth, then clearly we should revise the analysis as it provides us with a non-consistent framework, where our modeling is at odds with our data analysis. Furthermore, testing the methods on a simplified model is still advantageous compared with the option of using the methods without any validation. To mitigate the risk of using irrelevant models, in some cases it may be desirable to test the analysis methods on as broad a class of models as possible as we have done in the paper, for example by our use of a general value of $\alpha$ to describe the size-control strategy within our models. Thus, guided by the model, the data analysis methods can be ultimately applied to experimental data and underlying functional relationships can be inferred. Reiterating the message of the authors in *Hogg et al., 2010*, the data analysis using this framework aims to justify the methods being used, thus, reducing arbitrariness and promoting consensus among the scientists working in the field.

# Materials and methods
## Experimental methods

Strain engineering: STK13 strain (ΔftsN::frt-Ypet-FtsN, ΔdnaN::frt-mCherry-dnaN) is derivative of *E. coli* K12 BW27783 (CGSC#: 12119) constructed by $\lambda$-Red engineering (*Datsenko and Wanner, 2000*) and by P1 transduction (*Thom, 2007*). For chromosomal replacement of ftsN with fluorescence derivative, we used primers carrying 40nt tails with identical sequence to the *ftsN* chromosomal locus and a plasmid carrying a copy of *ypet* preceded by a kanamycin resistance cassette flanked by *frt* sites (frt-*kan^R*-frt-Ypet-linker) as PCR template (a kind gift from R. Reyes-Lamothe McGill University, Canada; *Reyes-Lamothe et al., 2010*). The resulting PCR product was transformed by electroporation into a strain carrying the $\lambda$-Red-expressing plasmid pKD46. Colonies were selected by kanamycin resistance, verified by fluorescence microscopy and by PCR using primers annealing to regions flanking ftsN gene. After removal of kanamycin resistance by expressing the Flp recombinase from plasmid pCP20 (*Cherepanov and Wackernagel, 1995*), we transferred the mCherry-dnaN gene fusion (BN1682 strain; a kind gift from Nynke Dekker from TUDelft, The Netherlands, *Moolman et al., 2014*) into the strain by P1 transduction. To minimize the effect of the insertion on the expression levels of the gene we removed the kanamycin cassette using Flp recombinase expressing plasmid pCP20.

Cells growth, preparation, and culturing *E. coli* in mother machine microfluidic devices: All cells were grown and imaged in M9 minimal medium (Teknova) supplemented with 2 mM magnesium sulfate (Sigma) and corresponding carbon sources at 28 °C. Three different carbon sources were used: 0.5 % glucose supplemented by 0.2 % casamino acids (Cas) (Sigma), 0.3 % glycerol (Fisher), and 0.3 % alanine (Fisher) supplemented with 1 x trace elements (Teknova).

For microscopy, we used mother machine microfluidic devices made of PDMS (polydimethylsiloxane). These were fabricated following to previously described procedure (*Yang et al., 2018*). To grow and image cells in microfluidic device, we pipetted 2–3 µl of resuspended concentrated overnight culture of $OD_{600}\sim$ 0.1 into main flow channel of the device and let cells to populate the dead-end channels. Once these channels were sufficiently populated (about 1 hr), tubing was connected to the device, and the flow of fresh M9 medium with BSA (0.75 µg/ml) was started. The flow was maintained at 5 µl/min during the entire experiment by an NE-1000 Syringe Pump (New Era Pump Systems, NY). To ensure steady-state growth, the cells were left to grow in channels for at least 14 hr before imaging started.

Microscopy: A Nikon Ti-E inverted epifluorescence microscope (Nikon Instruments, Japan) with a 100 X (NA = 1.45) oil immersion phase contrast objective (Nikon Instruments, Japan), was used for imaging the bacteria. Images were captured on an iXon DU897 EMCCD camera (Andor Technology, Ireland) and recorded using NIS-Elements software (Nikon Instruments, Japan). Fluorophores were excited by a 200 W Hg lamp through an ND8 neutral density filter. A Chroma 41,004 filtercube was used for capturing mCherry images, and a Chroma 41,001 (Chroma Technology Corp., VT) for Ypet images. A motorized stage and a perfect focus system were utilized throughout time-lapse imaging. Images in all growth conditions were obtained at 4 min frame rate.

Image analysis: Image analysis was carried out using Matlab (MathWorks, MA) scripts based on Matlab Image Analysis Toolbox, Optimization Toolbox, and DipImage Toolbox (https://www.diplib.org/). Cell lengths were determined based on segmented phase contrast images. Dissociation of Ypet-FtsN label from cell middle was used to determine the exact timing of cell divisions.

Further experimental details can also be found in *Tiruvadi-Krishnan et al., 2021*.

## Model

Consider a model of cell cycle characterized by two events: cell birth and division. In our model, we assume that, barring the noise, cells tend to divide at a particular size $v_d$ given size at birth $v_b$, via some regulatory mechanism. Hence, we can write $v_d$ as a function of $v_b$, $f(v_b)$. *Amir, 2014* provides a framework to capture the regulatory mechanisms by choosing $f(v_b) = 2\,v_b^{1-\alpha} v_0^{\alpha}$. $v_0$ is the typical size at birth and $\alpha$ captures the strength of regulation strategy. $\alpha = 0$ corresponds to the timer model where division occurs after a constant time from birth, and $\alpha = 1$ is the sizer where a cell divides on reaching a critical size. $\alpha = 1/2$ can be shown to be equivalent to an adder where division is controlled by addition of constant size from birth (*Amir, 2014*). From here on, we would be using the length of the cell ($L_b$, $L_d$, etc.) as a proxy for size ($v_b$, $v_d$, etc.). To reiterate, the length growth is not the same as cell volume growth as shown in Appendix 1. All of the variable definitions are summarized in *Supplementary file 1*. We also define $l_b = \frac{L_b}{\langle L_b \rangle}$ and $l_d = \frac{L_d}{\langle L_b \rangle}$. Using this, we can write the division strategy $f(l_b)$ to be $l_d = f(l_b) = 2\,l_b^{1-\alpha}$. The total division size obtained will be a combination of $f(l_b)$ and noise in the division timing, the source of which could be the stochasticity in biochemical reactions controlling division.

We will assume that division is perfectly symmetric i.e., size at birth in the $(n+1)^{th}$ generation ($l_b^{n+1}$) is half of size at division in the $n^{th}$ generation ($l_d^n$). Using the size additive division timing noise ($\zeta_s(0, \sigma_{bd})$) and $f(l_b)$ specified above, we obtain,

$$x_{n+1} = (1-\alpha)x_n + \ln\left(1 + \frac{\zeta_s(0,\sigma_{bd})}{2(1+x_n)^{1-\alpha}}\right), \qquad (4)$$

where $x_n = \ln(l_b^n)$. Size at birth ($L_b$) is narrowly distributed, hence $l_b \approx 1$ and we can write $x = \ln(l_b) = \ln(1+\delta)$ where $\delta$ is a small number. We obtain $x \ll 1$ and,

$$x \approx \delta = l_b - 1. \qquad (5)$$

The size additive noise, $\zeta_s(0, \sigma_{bd})$ is assumed to be small and has a normal distribution with mean 0 and standard deviation $\sigma_{bd}$. Note that $\sigma_{bd}$ is a dimensionless quantity. Since $\zeta_s(0, \sigma_{bd})$ is assumed to be small and $x_n \ll 1$, we can Taylor expand the last term of *Equation 4* to first order,

$$x_{n+1} \approx (1-\alpha)x_n + \frac{\zeta_s(0,\sigma_{bd})}{2}. \qquad (6)$$

*Equation 6* shows a recursive relation for cell size and it is agnostic of the mode of growth. We will show later for exponential growth that replacing the size additive noise with time additive noise does not change the structure of *Equation 6*.

## Exponential growth

Next, we will try to obtain the generation time ($T_d$) in the case of exponentially growing cells. For exponential growth, the time at division $T_d$ is given by,

$$T_d = \frac{1}{\lambda} \ln(\frac{L_d}{L_b}). \tag{7}$$

For simplicity, we will assume a constant growth rate ($\lambda$) within the cell-cycle. Growth rate is fixed at the start of the cell-cycle and is given by $\lambda = \langle\lambda\rangle + \langle\lambda\rangle\xi(0, CV_\lambda)$, where $\langle\lambda\rangle$ is the mean growth rate and $\xi(0, CV_\lambda)$ is assumed to be small with a normal distribution that has mean 0 and standard deviation $CV_\lambda$. $CV_\lambda$ denotes the coefficient of variation (CV) of the growth rate. This captures the variability in growth rate within cells arising from the stochastic nature of biochemical reactions occurring within the cell.

### Size additive noise

Here we will calculate the generation time using the division strategy f($l_b$) and a size additive division timing noise ($\zeta_s(0, \sigma_{bd})$) as described previously. On substituting $L_d = (f(l_b) + \zeta_s)\langle L_b\rangle$ into *Equation 7* we obtain,

$$T_d = \frac{1}{\langle\lambda\rangle + \langle\lambda\rangle\xi(0, CV_\lambda)} \ln(\frac{2l_b^{1-\alpha} + \zeta_s(0, \sigma_{bd})}{l_b}), \tag{8}$$

where the size additive noise ($\zeta_s(0, \sigma_{bd})$) is Gaussian with mean 0 and standard deviation $\sigma_{bd}$. The noise $\zeta_s(0, \sigma_{bd})$ is assumed to be small, and we obtain to first order,

$$T_d \approx \frac{1}{\lambda} \left( \ln(2) - \alpha x_n + \frac{\zeta_s(0, \sigma_{bd})}{2(1 + x_n)^{1-\alpha}} \right). \tag{9}$$

Since $x_n \ll 0$, on Taylor expanding $\frac{1}{(1 + x_n)^{1-\alpha}}$ to first order,

$$T_d \approx \frac{1}{\lambda} \left( \ln(2) - \alpha x_n + \frac{\zeta_s(0, \sigma_{bd})}{2}(1 + (1 - \alpha)x_n) \right). \tag{10}$$

Assuming noise in growth rate to be small and expanding to first order, we obtain,

$$T_d \approx \frac{1}{\langle\lambda\rangle} \left( \ln(2) - \alpha x_n - \ln(2)\xi(0, CV_\lambda) + \frac{\zeta_s(0, \sigma_{bd})}{2} \right). \tag{11}$$

*Equation 11* gives the generation time for the class of models where birth controls division under the assumption that growth is exponential.

### Time additive noise

Next, we ensure that the recursive relation for size at birth and the expression for the generation time given by *Equations 6 and 11*, respectively, are robust to the nature of noise assumed. In this section, the generation time is obtained using the division strategy f($l_b$) as described previously along with a time additive division timing noise ($\frac{\zeta}{\langle\lambda\rangle}$). In such a case, $T_d$ is obtained to be,

$$T_d = \frac{1}{\lambda}(\ln(2) - \alpha x_n) + \frac{\zeta(0, \sigma_n)}{\langle\lambda\rangle}. \tag{12}$$

The time additive noise, $\frac{\zeta(0, \sigma_n)}{\langle\lambda\rangle}$, is assumed to be small and has a normal distribution with mean 0 and standard deviation $\frac{\sigma_n}{\langle\lambda\rangle}$. Note that $\sigma_n$ is a dimensionless quantity.

Assuming noise in growth rate to be small, we find $T_d$ to first order to be,

$$T_d \approx \frac{1}{\langle\lambda\rangle} \left( \ln(2) - \alpha x_n - \ln(2)\xi(0, CV_\lambda) + \zeta(0, \sigma_n) \right). \tag{13}$$

*Equation 13* is same as *Equation 11*, if the time additive noise term, $\zeta(0, \sigma_n)$, in *Equation 12* is replaced by $\zeta_s(0, \sigma_{bd})/2$. Using *Equation 13*, the variance in $T_d$ ($\sigma_t^2$) is,

$$\sigma_t^2 = \frac{1}{\langle\lambda\rangle^2}\left(\ln^2(2)CV_\lambda^2 + \frac{2\sigma_n^2}{2-\alpha}\right). \tag{14}$$

For exponential growth, we also find,

$$\ln\left(\frac{L_d}{L_b}\right) = x_{n+1} - x_n + \ln(2) = \lambda T_d. \tag{15}$$

On substituting *Equation 12* into *Equation 15* we obtain to first order,

$$x_{n+1} \approx (1-\alpha)x_n + \zeta(0, \sigma_n). \tag{16}$$

On replacing the time additive noise term, $\zeta(0, \sigma_n)$, in *Equation 16* with $\zeta_s(0, \sigma_{bd})/2$, we recover the recursive relation for size at birth obtained in the case of size additive noise shown in *Equation 6*. Hence, the model is insensitive to noise being size additive or time additive with a simple mapping for going from one noise type to another in the small noise limit.

At steady state, $x$ has a normal distribution with mean 0 and variance $\sigma_x^2$ whose value is given by,

$$\sigma_x^2 = \frac{\sigma_n^2}{\alpha(2-\alpha)}. \tag{17}$$

We note that some of the derivations above have also been presented in *Eun et al., 2018*, but are provided here for completeness.

## Predicting the results of statistical constructs applied on $\ln(\frac{L_d}{L_b})$ vs $\langle\lambda\rangle T_d$ and $\langle\lambda\rangle T_d$ vs $\ln(\frac{L_d}{L_b})$

### Obtaining the best linear fit

Next, we calculate the equation for the best linear fit for the choice of $\ln(\frac{L_d}{L_b})$ as y-axis and $\langle\lambda\rangle T_d$ as x-axis and vice versa. For simplicity, in this section, we will consider time additive division timing noise. However, the results obtained here will hold for size additive noise as well because the model is robust to the type of noise added as shown in the previous section.

First, we calculate the correlation coefficient ($\rho_{exp}$) for $\ln(\frac{L_d}{L_b})$ and time of division $T_d$,

$$\rho_{exp} = \frac{\langle(\ln(\frac{L_d}{L_b}) - \langle\ln(\frac{L_d}{L_b})\rangle)(T_d - \langle T_d\rangle)\rangle}{\sigma_l\sigma_t}, \tag{18}$$

where $\sigma_l$ is the standard deviation in $\ln(\frac{L_d}{L_b})$. Using *Equations 15 and 16* we obtain,

$$\ln\left(\frac{L_d}{L_b}\right) \approx \ln(2) - \alpha x_n + \zeta(0, \sigma_n). \tag{19}$$

Substituting *Equations 13 and 19* into the numerator of *Equation 18*,

$$\langle(\ln\frac{L_d}{L_b}) - \langle\ln(\frac{L_d}{L_b})\rangle)(T_d - \langle T_d\rangle)\rangle$$
$$= \langle(-\alpha x_n + \zeta(0, \sigma_n))\frac{(-\alpha x_n - \ln(2)\xi(0, CV_\lambda) + \zeta(0, \sigma_n))}{\langle\lambda\rangle}\rangle. \tag{20}$$

As the terms $\zeta(0, \sigma_n)$, $\xi(0, CV_\lambda)$ and $x_n$ are independent of each other, $\langle\xi(0, CV_\lambda)\zeta(0, \sigma_n)\rangle = 0$, $\langle\xi(0, CV_\lambda)x_n\rangle = 0$ and $\langle x_n\zeta(0, \sigma_n)\rangle = 0$. *Equation 20* simplifies to,

$$\langle(\ln(\frac{L_d}{L_b}) - \langle\ln(\frac{L_d}{L_b})\rangle)(T_d - \langle T_d\rangle)\rangle = (\alpha^2\sigma_x^2 + \sigma_n^2)\frac{1}{\langle\lambda\rangle}. \tag{21}$$

The variance of $\ln(\frac{L_d}{L_b})$ obtained using *Equation 19* is,

$$\sigma_l^2 = \alpha^2\sigma_x^2 + \sigma_n^2 = \frac{2\sigma_n^2}{2-\alpha}. \tag{22}$$

Inserting *Equations 14, 21 and 22* into *Equation 18*, we get,

$$\rho_{exp} = \sqrt{\frac{1}{1 + \frac{(1 - \frac{\alpha}{2}) \ln^2(2) CV_\lambda^2}{\sigma_n^2}}}.$$ (23)

The slope of a linear regression line is given by,

$$m = \rho \frac{\sigma_y}{\sigma_x},$$ (24)

where $\sigma_x$, $\sigma_y$, and $\rho$ are the standard deviation of the x-variable, the standard deviation of the y-variable and the correlation coefficient of the (x,y) pair, respectively. The intercept is,

$$c = \langle y \rangle - m \langle x \rangle.$$ (25)

On the x-axis, we plot $\langle \lambda \rangle T_d$ and the y-axis is chosen as $\ln(\frac{L_d}{L_b})$. The slope for this choice ($m_{tl}$) can be calculated by,

$$m_{tl} = \rho_{exp} \frac{\sigma_l}{\sigma_t \langle \lambda \rangle}.$$ (26)

On substituting the values we get,

$$m_{tl} = \frac{1}{1 + \frac{(1 - \frac{\alpha}{2}) \ln^2(2) CV_\lambda^2}{\sigma_n^2}}.$$ (27)

Only for $CV_\lambda \ll \sigma_n$ we would expect a slope close to 1.
The intercept ($c_{tl}$) for the $\ln(\frac{L_d}{L_b})$ vs $\langle \lambda \rangle T_d$ plot is given by,

$$c_{tl} = \langle \ln(\frac{L_d}{L_b}) \rangle - m_{tl} \langle \langle \lambda \rangle T_d \rangle = \ln(2) \left( 1 - \frac{1}{1 + \frac{(1 - \frac{\alpha}{2}) \ln^2(2) CV_\lambda^2}{\sigma_n^2}} \right).$$ (28)

However, if we choose the x-axis as $\ln(\frac{L_d}{L_b})$ and the y-axis is chosen as $\langle \lambda \rangle T_d$, we obtain the slope $m_{lt}$,

$$m_{lt} = \rho_{exp} \frac{\sigma_t \langle \lambda \rangle}{\sigma_l}.$$ (29)

On substituting the values we obtain $m_{lt}$ = 1 independent of the noise parameters and find that the intercept is zero.

## Non-linearity in binned data

In the Main text, for the plot $\ln(\frac{L_d}{L_b})$ vs $\langle \lambda \rangle T_d$, we find the binned data to be non-linear (see **Figure 2C** of the Main text). In this section, we explain the non-linearity observed using the model developed in the previous sections.

Binning data based on the x-axis means taking an average of the y-variable conditioned on the value of the x-variable. Mathematically, this amounts to calculating $\mathbb{E}[y \mid x]$ i.e., the conditional expectation of the y-variable given that x is fixed. In our case, we need to calculate $\mathbb{E}[\ln(\frac{L_d}{L_b}) \mid \langle \lambda \rangle T_d]$. $\ln(\frac{L_d}{L_b}) = \lambda T_d$ by definition of exponential growth, hence,

$$\mathbb{E}[\ln(\frac{L_d}{L_b}) \mid \langle \lambda \rangle T_d] = \mathbb{E}[\lambda T_d \mid \langle \lambda \rangle T_d].$$ (30)

Since $T_d$ is fixed, this is equivalent to calculating $\mathbb{E}[\lambda \mid T_d]$. Using **Equation 13**,

$$\mathbb{E}[\lambda \mid T_d] = \frac{\int_{-\infty}^{\infty} \int_{-\infty}^{\infty} \int_{-\infty}^{\infty} \lambda p(x, \xi, \zeta) \, \delta(T_d - (\frac{\ln(2)}{\langle \lambda \rangle} - \alpha \frac{x}{\langle \lambda \rangle} - \frac{\ln(2)\xi}{\langle \lambda \rangle} + \frac{\zeta}{\langle \lambda \rangle})) \, dx \, d\xi \, d\zeta}{\int_{-\infty}^{\infty} \int_{-\infty}^{\infty} \int_{-\infty}^{\infty} p(x, \xi, \zeta) \, \delta(T_d - (\frac{\ln(2)}{\langle \lambda \rangle} - \alpha \frac{x}{\langle \lambda \rangle} - \frac{\ln(2)\xi}{\langle \lambda \rangle} + \frac{\zeta}{\langle \lambda \rangle})) \, dx \, d\xi \, d\zeta}.$$ (31)

$p(x, \xi, \zeta)$ is the joint probability distribution of $x$ and noise parameters $\xi$ and $\zeta$. Since, they are independent of each other, the joint distribution is product of the individual distributions $f_1(x)$, $f_2(\xi)$ and $f_3(\zeta)$, the distributions being Gaussian with mean 0 and standard deviation $\sigma_x$, $CV_\lambda$ and $\sigma_n$, respectively. $\sigma_x$, $\sigma_n$ are related by **Equation 17**. Since $x$, $\xi$, and $\zeta$ are narrowly distributed around zero, the contribution from large positive or negative values is extremely small. This ensures that $T_d$ is also

close to its mean and non-negative despite the limits of the integral being $-\infty$ to $\infty$. Using $\lambda = \langle\lambda\rangle + \langle\lambda\rangle\xi(0, CV_\lambda)$ in *Equation 31*,

$$\mathbb{E}[\lambda \mid T_d] = \langle\lambda\rangle \left(1 + \frac{\int_{-\infty}^{\infty}\int_{-\infty}^{\infty}\int_{-\infty}^{\infty} \xi f_1(x) f_2(\xi) f_3(\zeta)\, \delta(T_d - (\frac{\ln(2)}{\langle\lambda\rangle} - \alpha\frac{x}{\langle\lambda\rangle} - \frac{\ln(2)\xi}{\langle\lambda\rangle} + \frac{\zeta}{\langle\lambda\rangle}))\, dx\, d\xi\, d\zeta}{\int_{-\infty}^{\infty}\int_{-\infty}^{\infty}\int_{-\infty}^{\infty} f_1(x) f_2(\xi) f_3(\zeta)\, \delta(T_d - (\frac{\ln(2)}{\langle\lambda\rangle} - \alpha\frac{x}{\langle\lambda\rangle} - \frac{\ln(2)\xi}{\langle\lambda\rangle} + \frac{\zeta}{\langle\lambda\rangle}))\, dx\, d\xi\, d\zeta}\right).$$

(32)

On evaluating the integrals, we obtain,

$$\mathbb{E}[\lambda \mid T_d] = \langle\lambda\rangle \left(1 + \frac{1}{1 + \frac{2}{2-\alpha}\frac{\sigma_n^2}{CV_\lambda^2 \ln^2(2)}} - \frac{\frac{\langle\lambda\rangle T_d}{\ln(2)}}{1 + \frac{2}{2-\alpha}\frac{\sigma_n^2}{CV_\lambda^2 \ln^2(2)}}\right).$$

(33)

Thus, the trend of binned data is found to be,

$$\mathbb{E}[\ln(\frac{L_d}{L_b}) \mid \langle\lambda\rangle T_d] = \langle\lambda\rangle T_d \left(1 + \frac{1}{1 + \frac{2}{2-\alpha}\frac{\sigma_n^2}{CV_\lambda^2 \ln^2(2)}} - \frac{\frac{\langle\lambda\rangle T_d}{\ln(2)}}{1 + \frac{2}{2-\alpha}\frac{\sigma_n^2}{CV_\lambda^2 \ln^2(2)}}\right).$$

(34)

In the regime $CV_\lambda \ll \sigma_n$, the last two terms on the RHS of *Equation 34* vanish and the binned data follows the trend y = x.

For the $\langle\lambda\rangle T_d$ vs $\ln(\frac{L_d}{L_b})$ plot, we need to calculate $\mathbb{E}[\langle\lambda\rangle T_d \mid \ln(\frac{L_d}{L_b})]$. Using *Equations 13 and 19*, we obtain,

$$\langle\lambda\rangle T_d = \ln(\frac{L_d}{L_b}) - \ln(2)\xi(0, CV_\lambda).$$

(35)

$\ln(\frac{L_d}{L_b})$ is independent of $\xi(0, CV_\lambda)$. Using this, we can write $\mathbb{E}[\langle\lambda\rangle T_d \mid \ln(\frac{L_d}{L_b})]$ as,

$$\mathbb{E}[\langle\lambda\rangle T_d \mid \ln(\frac{L_d}{L_b})] = \frac{\int_{-\infty}^{\infty}\int_{-\infty}^{\infty} (\langle\lambda\rangle T_d)\, f_2(\xi)\, f_4(\ln(\frac{L_d}{L_b}))\, \delta\left(\langle\lambda\rangle T_d - (\ln(\frac{L_d}{L_b}) - \ln(2)\xi)\right)\, d(\langle\lambda\rangle T_d)\, d\xi}{f_4(\ln(\frac{L_d}{L_b}))}.$$

(36)

Note that the integral over $\langle\lambda\rangle T_d$ goes from $-\infty$ to $\infty$ although $\langle\lambda\rangle T_d$ cannot be negative. As before, this is not an issue because we assume $\langle\lambda\rangle T_d$ to be tightly regulated around $\ln(2)$ and the contribution to the integral from $-\infty$ to $0$ is negligible. $f_4(\ln(\frac{L_d}{L_b}))$ denotes the probability distribution for $\ln(\frac{L_d}{L_b})$, the distribution being Gaussian with mean $\ln(2)$, and standard deviation $\sigma_l$ which is calculated in *Equation 22*. Putting the Gaussian form of $f_2(\xi)$ into the integral and simplifying we get,

$$\mathbb{E}[\langle\lambda\rangle T_d \mid \ln(\frac{L_d}{L_b})] = \ln(\frac{L_d}{L_b}).$$

(37)

The trend of binned data to first order in noise and $x$ is $\mathbb{E}[\langle\lambda\rangle T_d \mid \ln(\frac{L_d}{L_b})] = \ln(\frac{L_d}{L_b})$. This is shown in *Figure 2D* of the Main text where the binned data follows the y = x line.

## Linear growth

In this section, we will focus on finding the equation of the best linear fit for relevant plots in the case of linear growth. The time at division for linear growth is given by,

$$T_d = \frac{L_d - L_b}{\lambda'}.$$

(38)

Note that $\lambda'$ has units of [length/time] and is defined as the elongation speed. This is different from the exponential growth rate which has units [1/time]. Here, we will work with the normalized length at birth ($l_b$) and division ($l_d$),

$$T_d = \frac{l_d - l_b}{\lambda_{lin}}.$$

(39)

Consider the normalized elongation speed to be $\lambda_{lin} = \langle\lambda_{lin}\rangle + \langle\lambda_{lin}\rangle\xi_{lin}(0, CV_{\lambda,lin})$, where $\langle\lambda_{lin}\rangle$ is the mean normalized elongation speed for a lineage of cells and $\xi_{lin}(0, CV_{\lambda,lin})$ is normally distributed with mean 0 and standard deviation $CV_{\lambda,lin}$. Thus, the CV of elongation speed is $CV_{\lambda,lin}$. The regulation

strategy which the cell undertakes is equivalent to that in previous sections and is given by $g(l_b) = 2 + 2(1 - \alpha)(l_b - 1)$. Note that we can obtain $g(l_b)$ by Taylor expanding $f(l_b)$ around $l_b = 1$. Using the regulation strategy $g(l_b)$ and adding a size additive noise $\zeta_s(0, \sigma_{bd})$ which is independent of $l_b$, we find,

$$T_d = \frac{2+2(1-\alpha)(l_b^n-1)+\zeta_s(0,\sigma_{bd})-l_b^n}{\langle\lambda_{lin}\rangle(1+\xi_{lin}(0,CV_{\lambda,lin}))}. \tag{40}$$

Note that we chose size additive division timing noise ($\zeta_s(0, \sigma_{bd})$) for convenience in this section. However, it can be shown as done previously that the model is robust to the noise in division timing being size additive or time additive. Assuming that the noise terms $\xi_{lin}(0, CV_{\lambda,lin})$ and $\zeta_s(0, \sigma_{bd})$ are small, we obtain to first order,

$$T_d \approx \frac{(1-2\alpha)(l_b-1)+1+\zeta_s(0,\sigma_{bd})-\xi_{lin}(0,CV_{\lambda,lin})}{\langle\lambda_{lin}\rangle}. \tag{41}$$

The terms $l_b$, $\zeta_s(0, \sigma_{bd})$ and $\xi_{lin}(0, CV_{\lambda,lin})$ are independent of each other. The standard deviation of $T_d$ ($\sigma_t$) can be calculated to be,

$$\sigma_t^2 = \frac{(1-2\alpha)^2\sigma_b^2+\sigma_{bd}^2+CV_{\lambda,lin}^2}{\langle\lambda_{lin}\rangle^2}. \tag{42}$$

Assuming perfectly symmetric division and using $l_d^n = g(l_b^n) + \zeta_s(0, \sigma_{bd})$, we find the recursive relation for $l_b^n$ to be,

$$l_d^n - l_b^n = 2l_b^{n+1} - l_b^n = (1 - 2\alpha)l_b^n + 2\alpha + \zeta_s(0, \sigma_{bd}). \tag{43}$$

Note that *Equation 43* is the same as *Equation 6* under the approximation $x_n = l_b^n - 1$. At steady state, the standard deviation of $l_b$ is denoted by $\sigma_b$ and using *Equation 43* its value is obtained to be,

$$\sigma_b^2 = \frac{\sigma_{bd}^2}{4\alpha(2-\alpha)}. \tag{44}$$

Similarly, the standard deviation of $l_d$-$l_b$, or equivalently $\lambda_{lin}T_d$, denoted by $\sigma_{l,lin}$, is calculated to be,

$$\sigma_{l,lin}^2 = \frac{4\alpha+1}{4\alpha(2-\alpha)}\sigma_{bd}^2. \tag{45}$$

For linear growth, a natural plot is $l_d$-$l_b$ vs $\langle\lambda_{lin}\rangle T_d$ (reminiscent of the $\ln(\frac{L_d}{L_b})$ vs $\langle\lambda\rangle T_d$ plot for exponential growth). To calculate the slope of the best linear fit, we have to calculate the correlation coefficient $\rho_{lin}$ given by,

$$\rho_{lin} = \frac{\langle(l_d-l_b-\langle l_d-l_b\rangle)\,(\langle\lambda_{lin}\rangle T_d-\langle\langle\lambda_{lin}\rangle T_d\rangle)\rangle}{\langle\lambda_{lin}\rangle\sigma_{l,lin}\sigma_t}. \tag{46}$$

Again using the independence of terms $l_b$, $\zeta_s(0, \sigma_{bd})$ and $\xi_{lin}(0, CV_{\lambda,lin})$ from each other, we get,

$$\rho_{lin} = \frac{(1-2\alpha)^2\sigma_b^2+\sigma_{bd}^2}{\langle\lambda_{lin}\rangle\sigma_{l,lin}\sigma_t} = \frac{\sigma_{l,lin}}{\langle\lambda_{lin}\rangle\sigma_t}. \tag{47}$$

The slope of best linear fit for the plot $l_d - l_b$ vs $\langle\lambda_{lin}\rangle T_d$ is given by,

$$m_{tl,lin} = \rho_{lin}\frac{\sigma_{l,lin}}{\langle\lambda_{lin}\rangle\sigma_t} = \frac{1}{1+\frac{CV_{\lambda,lin}^2}{\sigma_{bd}^2}\frac{4\alpha(2-\alpha)}{(4\alpha+1)}}. \tag{48}$$

The intercept $c_{tl,lin}$ is found to be,

$$c_{tl,lin} = \langle l_d - l_b\rangle - m_{tl,lin}\langle\langle\lambda_{lin}\rangle T_d\rangle = 1 - \frac{1}{1+\frac{CV_{\lambda,lin}^2}{\sigma_{bd}^2}\frac{4\alpha(2-\alpha)}{(4\alpha+1)}}. \tag{49}$$

On flipping the axis, the slope ($m_{lt,lin}$) for the plot $\langle\lambda_{lin}\rangle T_d$ vs $l_d - l_b$ is obtained to be,

$$m_{lt,lin} = \rho_{lin}\frac{\langle\lambda_{lin}\rangle\sigma_t}{\sigma_{l,lin}} = 1. \tag{50}$$

The intercept $c_{lt,lin}$ is found to be,

$$c_{lt,lin} = \langle\langle\lambda_{lin}\rangle T_d\rangle - m_{lt,lin}\langle l_d - l_b\rangle = 0. \tag{51}$$

The best linear fit for the $\langle\lambda_{lin}\rangle T_d$ vs $l_d - l_b$ plot follows the trend y = x.

Simulations of the adder model for linearly growing cells were carried out. The deviation of the best linear fit for the $l_d - l_b$ vs $\langle\lambda_{lin}\rangle T_d$ plot from the y = x line is shown in **Figure 3—figure supplement 1A**, while in **Figure 3—figure supplement 1B**, the best linear fit for the plot $\langle\lambda_{lin}\rangle T_d$ vs $l_d - l_b$ is shown to agree with the y = x line.

## Differentiating linear from exponential growth

In this section, we explore the equation for the best linear fit of $\langle\lambda_{lin}\rangle T_d$ vs $l_d - l_b$ plot in the case of exponential growth and $\langle\lambda\rangle T_d$ vs $\ln(\frac{L_d}{L_b})$ plot for linear growth. Intuitively, we expect the best linear fit in both cases to deviate from the y = x line. In this section, we will calculate the best linear fit explicitly. Surprisingly, we will find that, in the case of linear growth, the best linear fit for the $\langle\lambda\rangle T_d$ vs $\ln(\frac{L_d}{L_b})$ plot follows the y = x line closely.

Let us begin with exponential growth with growth rate, $\lambda = \langle\lambda\rangle + \langle\lambda\rangle\xi(0, CV_\lambda)$ as defined previously. Again, $\xi(0, CV_\lambda)$ has a normal distribution with mean 0 and standard deviation $CV_\lambda$, it being the CV of the growth rate. The time at division is given by **Equation 7**. The average growth rate $\langle\lambda\rangle = \langle\frac{\ln(2)}{T_d}\rangle \approx \frac{\ln(2)}{\langle T_d\rangle}$. For exponential growth, we will plot $\langle\lambda_{lin}\rangle T_d$ vs $l_d - l_b$. As previously defined, $\langle\lambda_{lin}\rangle$ is the mean normalized elongation speed and $\langle\lambda_{lin}\rangle = \langle\frac{1}{T_d}\rangle \approx \frac{1}{\langle T_d\rangle}$. $\langle\lambda_{lin}\rangle$ is related to $\langle\lambda\rangle$ by,

$$\langle\lambda_{lin}\rangle = \frac{\langle\lambda\rangle}{\ln(2)}. \tag{52}$$

$l_d - l_b$ can be calculated by using the regulation strategy $f(l_b)$ introduced in the Model section and a normally distributed size additive noise $\zeta_s(0, \sigma_{bd})$. Note that we have chosen the noise in division timing to be size additive. However, the model is robust to the choice of type of noise as we showed in the Exponential growth section. Using **Equations 5** and **6** we obtain,

$$l_d^n - l_b^n \approx 1 + (1 - 2\alpha)x_n + \zeta_s(0, \sigma_{bd}). \tag{53}$$

Using **Equation 11**, $\langle\lambda_{lin}\rangle T_d$ is obtained to be,

$$\langle\lambda_{lin}\rangle T_d = 1 - \frac{\alpha x}{\ln(2)} - \xi(0, CV_\lambda) + \frac{\zeta_s(0, \sigma_{bd})}{2\ln(2)}. \tag{54}$$

To calculate the expression for $m_{lt,lin}$, the slope of the best linear fit for $\langle\lambda_{lin}\rangle T_d$ vs $l_d - l_b$ plot, we first calculate $\rho_{lin}$ given by **Equation 46**. The expression for $\sigma_{l,lin}$ (standard deviation of $l_d - l_b$) and $\sigma_t$ (standard deviation of $T_d$) are found to be,

$$\sigma_{l,lin}^2 = (1 - 2\alpha)^2\sigma_x^2 + \sigma_{bd}^2, \tag{55}$$

$$\sigma_t^2 = \frac{1}{\langle\lambda_{lin}\rangle^2}\left((\frac{\alpha\sigma_x}{\ln(2)})^2 + CV_\lambda^2 + (\frac{\sigma_{bd}}{2\ln(2)})^2\right). \tag{56}$$

$\sigma_x$ is related to $\sigma_n$ via **Equation 17**. In Exponential growth section, we also showed that $\sigma_n = \frac{\sigma_{bd}}{2}$. Using these, we can write,

$$\sigma_x^2 = \frac{\sigma_{bd}^2}{4\alpha(2-\alpha)}. \tag{57}$$

Now using the expressions for $\sigma_t$, $\sigma_{l,lin}$ and the fact that $x$, $\xi(0, CV_\lambda)$ and $\zeta_s(0, \sigma_{bd})$ are independent of each other, we get,

$$\rho_{lin} = \frac{\frac{(2\alpha-1)\alpha\sigma_x^2}{\ln(2)} + \frac{\sigma_{bd}^2}{2\ln(2)}}{\langle\lambda_{lin}\rangle\sigma_{l,lin}\sigma_t}. \tag{58}$$

For the plot $\langle\lambda_{lin}\rangle T_d$ vs $l_d - l_b$, the slope $m_{lt,lin}$ is given by,

$$m_{lt,lin} = \rho_{lin}\frac{\sigma_t\langle\lambda_{lin}\rangle}{\sigma_{l,lin}} = \frac{\frac{(2\alpha-1)\alpha\sigma_x^2}{\ln(2)} + \frac{\sigma_{bd}^2}{2\ln(2)}}{\sigma_{l,lin}^2}. \tag{59}$$

Inserting *Equation 55* into *Equation 59* and substituting $\sigma_x^2$ given by *Equation 57*, we obtain,

$$m_{lt,lin} = \frac{1}{\ln(2)}\frac{3\alpha}{4\alpha+1}. \tag{60}$$

The intercept $c_{lt,lin}$ is found to be,

$$c_{lt,lin} = \langle\langle\lambda_{lin}\rangle T_d\rangle - m_{lt,lin}\langle l_d - l_b\rangle = 1 - \frac{1}{\ln(2)}\frac{3\alpha}{4\alpha+1}. \tag{61}$$

For the adder model ($\alpha = \frac{1}{2}$), we get the value of slope $m_{lin,lt} = \frac{1}{2\ln(2)} \approx 0.7213$ and intercept $c_{lin,lt}$ = $1 - \frac{1}{2\ln(2)} \approx 0.279$. This is different from the best linear fit obtained for same regulatory mechanism controlling division in linearly growing cells where we found that the best linear fit follows the y = x line. Intuitively, we expect the best linear fit of $\langle\lambda_{lin}\rangle T_d$ vs $l_d - l_b$ plot to deviate from y = x line in the case of exponential growth. We showed analytically that for a class of models where birth controls division, it is indeed the case. This is also shown using simulations of the adder model in *Figure 3—figure supplement 1C*.

In the ' Obtaining the best linear fit' section, we found the best linear fit for $\langle\lambda\rangle T_d$ vs $\ln(\frac{L_d}{L_b})$ plot to follow the y = x line for exponentially growing cells where division is regulated by birth event via regulation strategy f($l_b$). Next, we calculate the equation for the best linear fit of $\langle\lambda\rangle T_d$ vs $\ln(\frac{L_d}{L_b})$ plot given that growth is linear. The model for division control will be same as that in the Linear growth section that is, the regulation strategy for division is given by g($l_b$) = 2 + 2(1 − α)($l_b$ − 1) which is also equivalent to f($l_b$). The linearly growing cells grow with elongation speed $\lambda_{lin} = \langle\lambda_{lin}\rangle(1 + \xi_{lin}(0, CV_{\lambda,lin}))$. As discussed before, $\xi_{lin}(0, CV_{\lambda,lin})$ has a normal distribution with mean 0 and standard deviation $CV_{\lambda,lin}$, it being the CV of the elongation speed. Using *Equations 5* and *6*, we get,

$$\ln(\frac{L_d}{L_b}) = \ln(2) - \alpha x^n + \frac{\zeta_s(0,\sigma_{bd})}{2}. \tag{62}$$

Using *Equations 5* and *52*, we obtain from *Equation 41*,

$$\langle\lambda\rangle T_d = \ln(2) + (1 - 2\alpha)\ln(2)x + \ln(2)\zeta_s(0, \sigma_{bd}) - \ln(2)\xi_{lin}(0, CV_{\lambda,lin}). \tag{63}$$

Since $x$, $\xi_{lin}(0, CV_{\lambda,lin})$ and $\zeta_s(0, \sigma_{bd})$ are uncorrelated, the standard deviation of $\ln(\frac{L_d}{L_b})$ and $T_d$ denoted by $\sigma_l$ and $\sigma_t$ respectively are calculated to be,

$$\sigma_l^2 = \alpha^2\sigma_x^2 + \frac{\sigma_{bd}^2}{4}, \tag{64}$$

$$\sigma_t^2 = \frac{\ln^2(2)}{\langle\lambda\rangle^2}((1 - 2\alpha)^2\sigma_x^2 + \sigma_{bd}^2 + CV_{\lambda,lin}^2). \tag{65}$$

We calculate the correlation coefficient for the pair $(\ln(\frac{L_d}{L_b}), \langle\lambda\rangle T_d)$. Since the correlation coefficient is unaffected by multiplying one of the variables with a positive constant, we can calculate the correlation coefficient for the pair $(\ln(\frac{L_d}{L_b}), T_d)$ or $\rho_{exp}$ as given by *Equation 18*. Using the independence of terms $x$, $\xi_{lin}(0, CV_{\lambda,lin})$ and $\zeta_s(0, \sigma_{bd})$,

$$\rho_{exp} = \frac{\ln(2)(\sigma_x^2(2\alpha-1)\alpha + \frac{\sigma_{bd}^2}{2})}{\langle\lambda\rangle\sigma_l\sigma_t}. \tag{66}$$

For the plot $\langle\lambda\rangle T_d$ vs $\ln(\frac{L_d}{L_b})$, the slope $m_{lt}$ of the best linear fit is given by,

$$m_{lt} = \rho_{exp}\frac{\sigma_t\langle\lambda\rangle}{\sigma_l} = \frac{\ln(2)(\sigma_x^2(2\alpha-1)\alpha + \frac{\sigma_{bd}^2}{2})}{\sigma_l^2}. \tag{67}$$

Inserting *Equation 64* into *Equation 67* and using *Equation 57*, we get,

$$m_{lt} = \frac{3}{2}\ln(2) \approx 1.0397. \tag{68}$$

Similarly the intercept ($c_{lt}$) for the plot $\langle\lambda\rangle T_d$ vs $\ln(\frac{L_d}{L_b})$ is found to be,

$$c_{lt} = \langle\langle\lambda\rangle T_d\rangle - m_{lt}\langle\ln(\frac{L_d}{L_b})\rangle = \ln(2)(1 - \frac{3}{2}\ln(2)) \approx -0.0275. \tag{69}$$

This is very close to y = x trend obtained for the same regulatory mechanism controlling division in exponentially growing cells (*Figure 3A*).

## Growth rate vs age and elongation speed vs age plots

In the previous sections, we found that binning and linear regression on the plot $\ln(\frac{L_d}{L_b})$ vs $\langle\lambda\rangle T_d$, and the plot obtained by interchanging the axes, were inadequate to identify the mode of growth. In this section, we try to validate the growth rate vs age plot as a method to elucidate the mode of growth.

In addition to cell size at birth and division and the generation time, cell size trajectories (cell size, $L$ vs time from birth, $t$) were obtained for multiple cell cycles. In our case, the cell size trajectories were collected either via simulations (in *Figure 3B*) or from experiments (for *Figure 4A-C*) at intervals of 4 min. Note that if the measurements were to be carried out at equal length intervals instead of time, the results discussed in the paper would still remain unchanged. For each trajectory, growth rate at time $t$ or age $\frac{t}{T_d}$ is calculated as $\frac{1}{L(t)}\frac{L(t+\Delta t)-L(t)}{\Delta t}$ where $\Delta t$ is the time between consecutive measurements. To obtain elongation speed vs age plots, the formula before needs to be replaced with $\frac{L(t+\Delta t)-L(t)}{\Delta t}$. The growth rate is interpolated to contain 200 points at equal intervals of time for each cell trajectory. The growth rate trends appear to be robust with regards to a different number of interpolated points (from 100 to 500 points). To obtain the growth rate trend as a function of cell age, we use the method previously applied in *Nordholt et al., 2020*. In this method, growth rate is binned based on age for each individual trajectory (50 bins) and the average growth rate is obtained in each of the bins. The binned data trend for growth rate vs age is then found by taking the average of the growth rate in each bin over all trajectories. Binning the growth rate for each trajectory ensures that each trajectory has an equal contribution to the final growth rate trend so as to avoid inspection bias. This step is especially important when data collected at equal intervals of time is analyzed. In such a case, cells with larger generation times have a greater number of measurements than cells with smaller generation times. Obtaining the growth rate trend without binning growth rate for each trajectory would have biased the binned data trend for the growth rate vs age plot to a smaller value because of over-representation by slower-growing cells (or equivalently cells with longer generation time). This bias toward lower growth rate values in the growth rate vs age plots is an instance of inspection bias.

In *Figure 4A-C*, we find the growth rate obtained from *E. coli* experiments to change within the cell cycle. In the two slower growth media (*Figure 4A and B*), the growth rate is found to increase with cell age while for the fastest growth media (*Figure 4C*) the growth rate follows a non-monotonic behaviour similar to that observed in *Nordholt et al., 2020* for *B. subtilis*. Abrupt changes in growth rate are reported at constriction in *Reshes et al., 2008*; *Banerjee et al., 2017*. We find that the growth rate changes start before constriction in the two slower growth conditions considered. One possibility is that this increase is due to preseptal cell wall synthesis (*Pazos et al., 2018*). Preseptal cell wall synthesis does not require activity of PBP3 (FtsI) but instead relies on bifunctional glycosyltransferases PBP1A and PBP1B that link to FtsZ via ZipA. One hypothesis that can be tested in future works is that at the onset of constriction, activity from PBP1A and PBP1B starts to gradually shift to the PBP3/FtsW complex and therefore no abrupt change in growth rate is observed. In the fastest growth condition (glucose-cas medium), we find that the increase in growth rate approximately coincides with onset of constriction, in agreement with the previous findings (*Reshes et al., 2008*; *Banerjee et al., 2017*).

In *Figure 4A-C*, the growth rate trends are not obtained for age close to one. This is because growth rate at age = 1 is given by $\frac{1}{L(T_d)}\frac{L(T_d+\Delta t)-L(T_d)}{\Delta t}$ and this requires knowing the cell lengths beyond the division event ($L(T_d+\Delta t)$). To estimate growth rates at age close to one, we approximate $L(T_d+\Delta t)$ to be the sum of cell sizes of the two daughter cells. In order to minimize inspection bias, we considered only those cell size trajectories which had $L(t)$ data for 12 min after division (corresponding to an age of approximately 1.1). However, the growth rate trends in all three growth media were robust with regards to a different time for which $L(t)$ was considered (4 min to 20 min after division). We use the binning procedure discussed before in this section. To validate this method, we applied it on synthetic data obtained from the simulations of exponentially growing cells following the adder and the adder per origin model. Cells were assumed to divide in a perfectly symmetric manner and both of the daughter cells were assumed to grow with the same growth rate, independent of the growth rate in the mother cell. The growth rate trends for the two models considered (adder and adder per origin) are expected to be constant even for cell age >1.

We found that the growth rate trends were indeed approximately constant as shown in *Figure 4—figure supplement 1D*. We also considered linear growth with division controlled via an adder model. The daughter cells were assumed to grow with the same elongation speed, independent of the elongation speed in the mother cell. In this case, we expect the elongation speed trend to be constant for cell age >1. This is indeed what we observed as shown in the inset of *Figure 4—figure supplement 1D*. We used this method on *E. coli* experimental data and found that the growth rate trends obtained for the three growth conditions (*Figure 4—figure supplement 1A–1C*) were consistent with that shown in *Figure 4A-C* in the relevant age ranges. For cell age close to one, we found that the growth rate decreased to a value close to the growth rate near cell birth (age ≈ 0) for all three growth conditions considered.

In summary, we find that the growth rate vs age plots are a consistent method to probe the mode of cell growth within a cell cycle.

## Growth rate vs time from specific event plots are affected by inspection bias

To probe the growth rate trend in relation to a specific cell cycle event, for example cell birth, growth rate vs time from birth plots are obtained for simulations of exponentially growing cells following the adder model. In the growth rate vs time from birth plot, the rate is found to stay constant and then decrease at longer times (*Figure 3—figure supplement 2C*) even though cells are exponentially growing. Because of inspection bias (or survivor bias), at later times, only the cells with larger generation times (or slower growth rates) 'survive'. The average generation time of the cells averaged upon in each bin of *Figure 3—figure supplement 2C* is shown in *Figure 3—figure supplement 2D*. The decrease in growth rate in *Figure 3—figure supplement 2C* occurs around the same time when an increase in generation time is observed in *Figure 3—figure supplement 2D*. Thus, the trend in growth rate is biased toward lower values at longer times. The problem might be circumvented by restricting the time on the x-axis to the smallest generation time of all the cell cycles considered (*Messelink et al., 2020*).

To check for growth rate changes at constriction, we used plots of growth rate vs time from constriction ($t − T_n$). Growth rate trends obtained from *E. coli* experimental data show a decrease at the edges of the plots (*Figure 4—figure supplement 2A and C*, and 2E). These deviate from the trends obtained using the growth rate vs age plots (*Figure 4A-C*). To investigate this discrepancy, we use a model which takes into account the constriction and the division event. Currently, it is unknown how constriction is related to division. For the purpose of methods validation, we use a model where cells grow exponentially, constriction occurs after a constant size addition from birth, and division occurs after a constant size addition from constriction. Note that other models where constriction occurs after a constant size addition from birth while division occurs after a constant time from constriction, as well as a mixed timer-adder model proposed in *Banerjee et al., 2017*, lead to similar results. We expect the growth rate trend to be constant for exponentially growing cells. However, we find using numerical simulations that it decreases at the plot edges both before and after the constriction event (*Figure 3—figure supplement 2A*). This decrease can be attributed to inspection bias. The average growth rate in time bins at the extremes are biased by cells with smaller growth rates. This is shown in *Figure 3—figure supplement 2B* where the average generation time for the cells contributing in each of the bins of *Figure 3—figure supplement 2A* is plotted. The time at which the growth rate decreases on both sides of the constriction event is close to the time at which the average generation time increases. For example, in alanine medium, the generation time for each of the bins is plotted in *Figure 4—figure supplement 2B*. The average generation time for the cells contributing to each of the bins is almost constant for the timings between –80 min and 20 min. Thus, for this time range the changes in growth rate are not because of inspection bias but are a real biological effect. The behavior of growth rate within this time range in *Figure 4—figure supplement 2A* is in agreement with the trend in growth rate vs age plot of *Figure 4A*. On accounting for inspection bias, the growth rate vs age plots agree with the growth rate vs time from constriction plots in other growth media as well (*Figure 4—figure supplement 2C*, *Figure 4—figure supplement 2E*). Thus, growth rate vs time plots are also a consistent method to probe growth rate modulation in the time range when avoiding the regimes prone to inspection bias.

## Results of elongation speed vs size plots are model-dependent

Cells assumed to undergo exponential growth have elongation speed proportional to their size. In the case of exponential growth, the binned data trend of the plot elongation speed vs size is expected to be linear with the slope of the best linear fit providing the value of growth rate and intercept being zero. In this section, we use the simulations to test if binning and linear regression on the elongation speed vs size plots are suitable methods to differentiate exponential growth from linear growth (*Cadart et al., 2019*).

To test the method, we generate cell size trajectories using simulations of the adder model with a size additive division timing noise and assuming exponential growth. Elongation speed at size $L(t)$ is calculated for each trajectory as $\frac{L(t+\Delta t)-L(t)}{\Delta t}$ where $\Delta t$ is the time between consecutive measurements ( = 4 min in our case). Each trajectory is binned into 10 equally sized bins based on their cell sizes and the average elongation speed is obtained for each bin. The final trend of elongation speed as a function of size is then obtained by binning (based on size) the pooled average elongation speed data of all the cell cycles.

We find that the binned data trend is linear with the slope of the best linear fit close to the average growth rate considered in the simulations (*Figure 3—figure supplement 3D*). This is in agreement with our expectations for exponential growth. In order to check if this method could differentiate between exponential growth and linear growth, we used simulations of the adder model undergoing linear growth to generate cell size trajectories for multiple cell cycles. For linear growth, elongation speed is expected to be constant, independent of its cell size. The binned data trend for the elongation speed vs size plot is also obtained to be constant for the simulations of linearly growing cells (*Figure 3—figure supplement 3B*). The intercept of the best linear fit obtained is close to the average elongation speed considered in the simulations. The binned data trend for linear and exponential growth are clearly different as shown in *Figure 3—figure supplement 3B* and *Figure 3—figure supplement 3D*, respectively, and this result holds for a broad class of models where the division event is controlled by birth and the growth rate (for exponential growth)/elongation speed (for linear growth) is distributed normally and independently between cell-cycles.

Next, we consider the adder per origin cell cycle model for exponentially growing cells (*Ho and Amir, 2015*). In this model space, the cell initiates DNA replication by adding a constant size per origin from the previous initiation size. The division occurs on average after a constant time from initiation. For exponentially growing cells, the binned data trend is still expected to be linear as before. Instead, we find using simulations that the trend is non-linear and it might be misinterpreted as non-exponential growth (*Figure 3—figure supplement 3F*).

Thus, the results of binning and linear regression for the plot elongation speed vs size is model-dependent.

## Interchanging axes in growth rate vs inverse generation time plot might lead to different interpretations

So far, our discussion was focused on the question of mode of single-cell growth. A related problem regards the relation between growth rate ($\lambda$) and the inverse generation time ($\frac{1}{T_d}$). On a population level, the two are clearly proportional to each other. However, single-cell studies based on binning showed an intriguing non-linear dependence between the two, with the two variables becoming uncorrelated in the faster-growth media (*Kennard et al., 2016*; *Iyer-Biswas et al., 2014*). Within the same medium, the binned data curve for the plot $\lambda$ vs $\frac{1}{T_d}$ flattened out for faster dividing cells. The trend in the binned data was different from the trend of y = ln(2) x line as observed for the population means. A priori one might speculate that the flattening in faster dividing cells could be because the faster dividing cells might have less time to adapt their division rate to transient fluctuations in the environment. *Kennard et al., 2016* insightfully also plotted $\frac{1}{T_d}$ vs $\lambda$ and found a collapse of the binned data for all growth conditions onto the y = $\frac{1}{\ln(2)}$ x line. These results are reminiscent of what we previously showed for the relation of $\ln(\frac{L_d}{L_b})$ and $\langle\lambda\rangle T_d$.

In the following, we will elucidate why this occurs in this case using an underlying model and predicting the trend based on it. We use simulations of the adder model undergoing exponential growth. The parameters for size added in a cell cycle and mean growth rates are extracted from the experimental data. CV of growth rate is assumed lower in faster growth media as observed by Kennard et al. Using this model, we could obtain the same pattern of flattening at faster growth

conditions that is observed in the experiments (**Figure 2—figure supplement 2A**). The population mean for $\lambda$ and $\frac{1}{T_d}$ follows the expected y = ln(2) x equation (shown as black dashed line) as was the case in experiments. Intuitively, such a departure from the expected y = ln(2) x line for the single-cell data can again be explained by determining the effect of noise on variables plotted on both axes. As previously stated $T_d$ is affected by both growth rate noise and noise in division timing while growth rate fluctuates independently of other sources of noise. This does not agree with the assumption for binning as noise in division timing affects the x-axis variable rather than the y-axis variable. In such a case, the trend in the binned data might not follow the expected y = ln(2) x line. However, on interchanging the axes, we would expect the assumptions of binning to be met and the trend to follow the y = $\frac{1}{\ln(2)}$ x line (**Figure 2—figure supplement 2B**).

## Data and simulations

### Experimental data

Experimental data obtained by **Tanouchi et al., 2017** was used to plot $L_d$ vs $L_b$ shown in **Figure 1A**. *E. coli* cells were grown at 25 °C in a mother machine device and the length at birth and division were collected for multiple cell cycles. $L_d$ vs $L_b$ plot was obtained using these cells and linear regression performed on it provided a best linear fit.

Data from recent mother machine experiments on *E. coli* was used to make all other plots. Details are provided in the Experimental methods and **Tiruvadi-Krishnan et al., 2021**. The experiments were conducted at 28 °C in three different growth conditions - alanine, glycerol, and glucose-cas (also see Experimental methods). Cell size trajectories were collected for multiple cell cycles and all of the data collected were considered while making the plots in the paper.

### Simulations

MATLAB R2021a was used for simulations. Simulations of the adder model for exponentially growing cells were carried out over a single lineage of 2500 generations (**Figure 2C and D**, **Figure 3—figure supplement 1C**). The mean length added between birth and division was set to 1.73 $\mu m$ in line with the experimental results for alanine medium. Growth rate was variable and sampled from a normal distribution at the start of each cell cycle. The mean growth rate was set to $\frac{\ln(2)}{\langle T_d \rangle}$, where $\langle T_d \rangle$ = 212 min and coefficient of variation (CV) = $CV_\lambda$ = 0.15. The noise in division timing was assumed to be time additive with mean 0 and standard deviation $\frac{\sigma_n}{\langle \lambda \rangle}$, where $\sigma_n$ = 0.15. The binning data trends and the best linear fits obtained using these simulations could be compared with the analytical results obtained in sections 'Non-linearity in binned data' and 'Differentiating linear from exponential growth'.

For simulations of linear growth (**Figure 3A-B**, **Figure 3—figure supplement 1A and B**, **Figure 3—figure supplement 3A, B** , **Figure 4—figure supplement 1D**), the mean growth rate was set to $\frac{\langle L_d - L_b \rangle}{\langle T_d \rangle}$, with the values of $\langle L_d - L_b \rangle$ and $\langle T_d \rangle$ used as mentioned previously. The noise in division timing was size additive with standard deviation = 0.15 $\langle L_b \rangle$. Noise was also considered to be size additive with the same standard deviation for the simulations of exponentially growing cells shown in **Figure 3B**, **Figure 3—figure supplement Figure 3—figure supplements 2C and 3C. D**, and **Figure 4—figure supplement 1D**.

In the simulations of super-exponential growth carried over a single lineage of 2500 generations (**Figure 3B**), the cells initially grew exponentially but in the later stages of the cell cycle, the growth rate increased as,

$$\frac{d\lambda}{dt} = 2k(t - t_c), \tag{70}$$

where k was fixed to be $\frac{2}{T_d^3}$ and $t_c$ was the time from birth at which the growth rate changed from exponential to super-exponential growth. $t_c$ was fixed to be half of the generation time of the cell or equivalently an age of 0.5. The division size was set by the adder model with a time additive noise with similar parameters as before for exponential growth simulations. The exponential growth rate at the start of each cell cycle was drawn from a normal distribution with mean set to $\frac{\ln(2)}{242} min^{-1}$ and CV = 0.15.

For **Figure 3B**, **Figure 3—figure supplement 3E and F**, **Figure 4—figure supplement 1D**, simulations were carried out over a lineage of 2500 generations for exponentially growing cells following the adder per origin model. In the simulations, the time increment is 0.01 min. The initial condition for the simulations is that cells are born and initiate DNA replication at time t = 0 but the results are

independent of initial conditions. The number of origins is also tracked throughout the simulations beginning with an initial value of 2. Cells divide into two daughter cells in a perfectly symmetrical manner (no noise in division ratio), and one of the daughter cells is discarded for the next cell cycle. In simulations, the growth rate was fixed within a cell cycle but varied between different cell cycles. On division, the growth rate for that cell cycle was drawn from a normal distribution with mean $\langle\lambda\rangle$ and coefficient of variation ($CV_\lambda$) whose values were fixed using the experimental data from alanine medium. The total length at which the next initiation happens is determined by,

$$L_i^{tot,next} = L_i + O\Delta_{ii}, \tag{71}$$

where $\Delta_{ii}$ is the length added per origin and O is the number of origins. To determine $L_i^{tot,next}$, $\Delta_{ii}$ was drawn on reaching initiation length from a normal distribution. The mean and CV of $\Delta_{ii}$ was obtained from experiments done in alanine medium. In the adder per origin model, division happens after a C + D time from initiation. The division length ($L_d$) is obtained to be,

$$L_d = L_i e^{\lambda(C+D)}. \tag{72}$$

In the simulations, once the initiation length was reached, the corresponding division occurred a time C + D after initiation. C + D timings for each initiation event were again drawn from a normal distribution with the same mean and CV as that of the experiments in alanine medium.

For *Figure 3—figure supplement 2A*, cells were assumed to grow exponentially in the simulations. The constriction length ($L_n$) was set to be,

$$L_n = L_b + \Delta_{bn}. \tag{73}$$

The length added ($\Delta_{bn}$) was assumed to have a normal distribution with the mean length added between birth and constriction set to 1.18 $\mu m$ and the CV = 0.23, in line with the experimental results for alanine medium. The length at division was set as,

$$L_d = L_n + \Delta_{nd}. \tag{74}$$

The length added ($\Delta_{nd}$) was also assumed to have a normal distribution with the mean length added set to 0.53 μm and the CV = 0.26, again in line with the experimental results for alanine medium.

For *Figure 3B*, *Figure 3—figure supplement 2A–2D*, 3 A-3F, *Figure 4—figure supplement 1D*, the cell sizes are recorded within the cell cycle at equal intervals of 4 min, similar to that in the *E. coli* experiments of *Tiruvadi-Krishnan et al., 2021*.

For simulations shown in *Figure 4—figure supplement 1D*, the cell size trajectories are obtained at intervals of 4 min beyond the current cell-cycle. The size after the division event is said to be the sum of the sizes of the daughter cells. It is also further assumed that the daughter cells are equal in size (perfectly symmetric division) and they both grow with the same growth rate (for exponential growth) or elongation speed (for linear growth). The growth rates/elongation speeds for the daughter cells are sampled from a normal distribution with a mean and CV as discussed before. The cell size trajectories are recorded for 80 min after the division event in the current cell cycle.

In *Figure 2—figure supplement 2*, simulations of the adder model for exponentially growing cells were carried out until a population of 5000 cells was reached. The parameters for size added in a cell cycle and mean growth rates were extracted from the experimental data (*Kennard et al., 2016*). The value of $\sigma_n$ used in all growth conditions was 0.17 while $CV_\lambda$ decreased in faster growth conditions (0.2 in the three slowest growth conditions, 0.12 and 0.07 in the second fastest and fastest growth conditions, respectively).

## Acknowledgements

The authors thank Ethan Levien, and Jie Lin for useful discussions, Jane Kondev, Xili Liu, and Marco Cosentino Lagomarsino for their useful feedback on the manuscript, Da Yang and Scott Retterer for help in microfluidic chip making, and Rodrigo Reyes-Lamothe for a kind gift of strain. Authors acknowledge technical assistance and material support from the Center for Environmental Biotechnology at the University of Tennessee. A part of this research was conducted at the Center for

Nanophase Materials Sciences, which is sponsored at Oak Ridge National Laboratory by the Scientific User Facilities Division, Office of Basic Energy Sciences, U.S. Department of Energy. This work has been supported by the US-Israel BSF research grant 2017004 (JM), the National Institutes of Health award under R01GM127413 (JM), NSF CAREER 1752024 (AA), NIH grant R01 AI143611 (AA) and NSF award 1806818 (PK).

## Additional information

### Funding

| Funder | Grant reference number | Author |
|---|---|---|
| US-Israel BSF Research Grant | 2017004 | Jaan Männik |
| National Institutes of Health | R01GM127413 | Jaan Männik |
| National Science Foundation | NSF CAREER 1752024 | Ariel Amir |
| National Science Foundation | NSF award 1806818 | Prathitha Kar |
| National Institutes of Health | R01 AI143611 | Ariel Amir |

The funders had no role in study design, data collection and interpretation, or the decision to submit the work for publication.

### Author contributions

Prathitha Kar, Conceptualization, Formal analysis, Methodology, Writing - original draft, Writing - review and editing; Sriram Tiruvadi-Krishnan, Jaana Männik, Conceptualization, Methodology, Writing - review and editing; Jaan Männik, Ariel Amir, Conceptualization, Formal analysis, Methodology, Writing - review and editing

### Author ORCIDs

Prathitha Kar (ID) http://orcid.org/0000-0002-4091-6860
Jaana Männik (ID) http://orcid.org/0000-0002-0777-7846
Jaan Männik (ID) http://orcid.org/0000-0002-6759-3053
Ariel Amir (ID) http://orcid.org/0000-0003-2611-0139

### Decision letter and Author response

Decision letter https://doi.org/10.7554/eLife.72565.sa1
Author response https://doi.org/10.7554/eLife.72565.sa2

## Additional files

### Supplementary files
- Transparent reporting form
- Supplementary file 1. Supplementary Information.

### Data availability

All data generated during this study are deposited in Dataverse (https://doi.org/10.7910/DVN/BNQUDW).

The following dataset was generated:

| Author(s) | Year | Dataset title | Dataset URL | Database and Identifier |
|---|---|---|---|---|
| Kar P, Tiruvadi-Krishnan S, Männik J, Männik J, Amir A | 2021 | Distinguishing different modes of growth using single-cell data | https://doi.org/10.7910/DVN/BNQUDW | Harvard Dataverse, 10.7910/DVN/BNQUDW |

The following previously published datasets were used:

| Author(s) | Year | Dataset title | Dataset URL | Database and Identifier |
|---|---|---|---|---|
| Tanouchi Y, Pai A, Park H, Huang S, Buchler NE, You L | 2017 | Data from long-term growth data of *Escherichia coli* at a single-cell level | https://doi.org/10.6084/m9.figshare.c.3493548.v1 | figshare, 10.6084/m9.figshare.c.3493548.v1 |

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

## Appendix 1

### Comparing length, surface area and volume growth rate

In the paper, we use cell length to represent cell size. However, other cell size characteristics such as cell surface area and cell volume could also be used to denote cell size. How does the growth rate vary with our choice of cell length, cell surface area, or cell volume to be the cell size?

To study this, we assume a cell morphology as shown in *Appendix 1—figure 1*. We assume that *E. coli* cells are cylindrical with hemispherical poles. The total length of the cell is $L$ with a radius $R$. The cell volume ($V$) is then,

$$V = \pi R^2 L - \frac{2}{3}\pi R^3. \tag{1-A1}$$

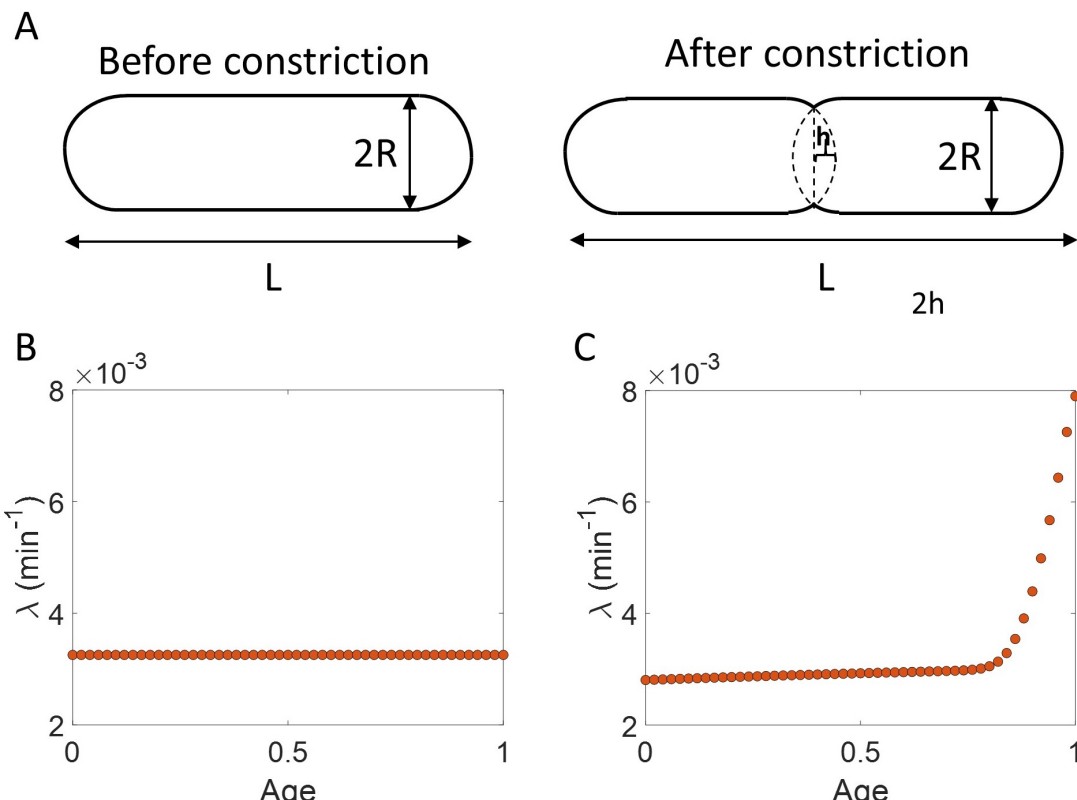

**Appendix 1—figure 1.** Length growth rate vs volume and surface area growth rate. (**A**) Cell morphology of *E. coli* used in the model is shown. The *E. coli* cells are assumed to be cylindrical with hemispherical end caps. Before constriction, the cell elongates with constant width (2 $R$). However, after onset of constriction, the septum starts forming at the mid-cell. (**B**) Length growth rate as a function of age assuming that the total cell surface area growth is exponential, and the radius is constant ($R = 0.35$ $\mu m$). (**C**) Length growth rate as a function of age assuming that the volume growth is exponential, radius is constant ($R = 0.35$ $\mu m$) and septum surface grows at a constant rate.

The morphology of the cell after constriction is also shown in *Appendix 1—figure 1A*. The volume in this case is,

$$V = \pi R^2 L - \frac{4}{3}\pi R^3 + 2\pi R^2 h - 2\pi h^2 R + \frac{2}{3}\pi h^3. \tag{2-A1}$$

If we make the assumption that cell biomass grows exponentially and the total cell surface area is coupled to the biomass (*Oldewurtel et al., 2021*), then cell surface area grows exponentially with time. Using the morphology in *Appendix 1—figure 1A*, the total surface area ($S$) before and after constriction is,

$$S = 2\pi RL. \tag{3-A1}$$

Surprisingly, this is independent of h. Since the surface area is proportional to the cell length (**Equation (3-A1)**), the length growth is also exponential with an identical growth rate as surface area growth, assuming the width of the cell is constant. The exponential growth of cell length is shown in **Appendix 1—figure 1B** using simulations where the cell surface is assumed to grow exponentially. So, for this model of cell growth and morphology, the length and the surface growth rates are found to be identical.

Next, we compare length growth rate to volume growth rate considering the same cell morphology as that in **Appendix 1—figure 1A**. In this model, the *volume* growth is assumed to be exponential. The volume before and after the onset of constriction are given by **Equations (1-A1)** and (2-A1), respectively.

Before constriction, the volume grows only by an increase in length of the cylindrical part of the cell while the width stays constant. However, after the constriction at mid-cell starts, the volume grows by an increase in length as well as by adding a septum surface at the mid-cell. We assume that the septum wall surface grows at a constant rate ($c_1$) (**Reshes et al., 2008**). We can obtain $c_1$ in terms of cell morphology variables to be,

$$c_1 = -4\pi R \frac{dh}{dt}. \tag{4-A1}$$

We can solve for $h(t)$ using the following boundary conditions,

$$h(t = T_n) = R, h(t = T_d) = 0, \tag{5-A1}$$

where $T_n$ is the time from birth at which constriction starts. Using **Equation (4-A1)** and **Equation (5-A1)**, we can obtain $c_1$ in terms of cell cycle variables $R$, $T_n$ and $T_d$,

$$c_1 = \frac{4\pi R^2}{T_d - T_n}. \tag{6-A1}$$

Under these assumptions, for exponential volume growth, we obtain the length growth via simulations. The length growth rate is shown in **Appendix 1—figure 1C**. The growth rate, the length at birth, the time at constriction from birth and the generation time parameters used in the simulations are obtained from experimental data in alanine growth medium. The width of the cells is assumed to be 0.35 $\mu m$. We find that before constriction, the length growth rate increases to a small extent ($\approx 6\%$). However, after constriction there is a rapid increase in length growth rate. Since the length growth rate increases with age, the length growth is super-exponential. Thus, the mode of growth in length and volume are not identical.

## Appendix 2

### Linear regression on $\ln(\frac{L_d}{L_b})$ vs $\langle T_d \rangle \lambda$ plot and its interchanged axes plot

In section 'Statistical methods like binning and linear regression should be interpreted based on a model', we found that binning and linear regression on the plots $\ln(\frac{L_d}{L_b})$ vs $\langle \lambda \rangle T_d$ and its interchanged axes were not a suitable method to identify the underlying mode of growth. In this section, we explore binning and linear regression on similar plots $\ln(\frac{L_d}{L_b})$ vs $\langle T_d \rangle \lambda$ plot and its interchanged axes. We test the usability of these plots to elucidate the mode of growth using the methodology proposed in the paper.

Assuming exponential growth, $\lambda$ for a cell cycle can be calculated as $\frac{1}{T_d} \ln(\frac{L_d}{L_b})$. On plotting $\ln(\frac{L_d}{L_b})$ vs $\langle T_d \rangle \lambda$ (*Appendix 2—figure 1*) and $\langle T_d \rangle \lambda$ vs $\ln(\frac{L_d}{L_b})$ (*Appendix 2—figure 1*) for the experimental data, we obtain the slope of the best linear fit to be close to zero (values shown in Table 1-Appendix 2). Next, using the methodology of the paper, we interpret these results using an underlying model. We consider a model in which cells grow exponentially with the division determined by birth. In the model, growth rate is fixed at the beginning of each cell cycle and is independent of size at birth. The model predicts that $\ln(\frac{L_d}{L_b})$ will be independent of the growth rate (*Equation 19* in main text). Thus, we would expect the slope to be zero for both of the plots $\ln(\frac{L_d}{L_b})$ vs $\langle T_d \rangle \lambda$ and $\langle T_d \rangle \lambda$ vs $\ln(\frac{L_d}{L_b})$). This is also shown using simulations of the adder model in *Appendix 2—figure 1* where the slope of the plots is close to zero. In order to differentiate between exponential growth and linear growth, the best linear fit in case of linear growth for these plots must deviate from y = constant line. However, we find for the simulations of the adder model where cells grow linearly that the slope of the best linear fit for both of the above plots is still zero (*Appendix 2—figure 1*). Note that $\lambda$ in the case of linear growth is still calculated as $\frac{1}{T_d} \ln(\frac{L_d}{L_b})$. A slope of zero in case of linear growth can be explained using *Equation 62* of the main text. Using the equation, we find that $\ln(\frac{L_d}{L_b})$ is independent of the underlying growth rate for linear growth. Thus, the best linear fit for both plots have a slope of zero in the case of linear growth. This indicates that binning and linear regression on the $\ln(\frac{L_d}{L_b})$ vs $\langle T_d \rangle \lambda$ and its interchanged axes plots are unsuitable for elucidating the mode of growth.

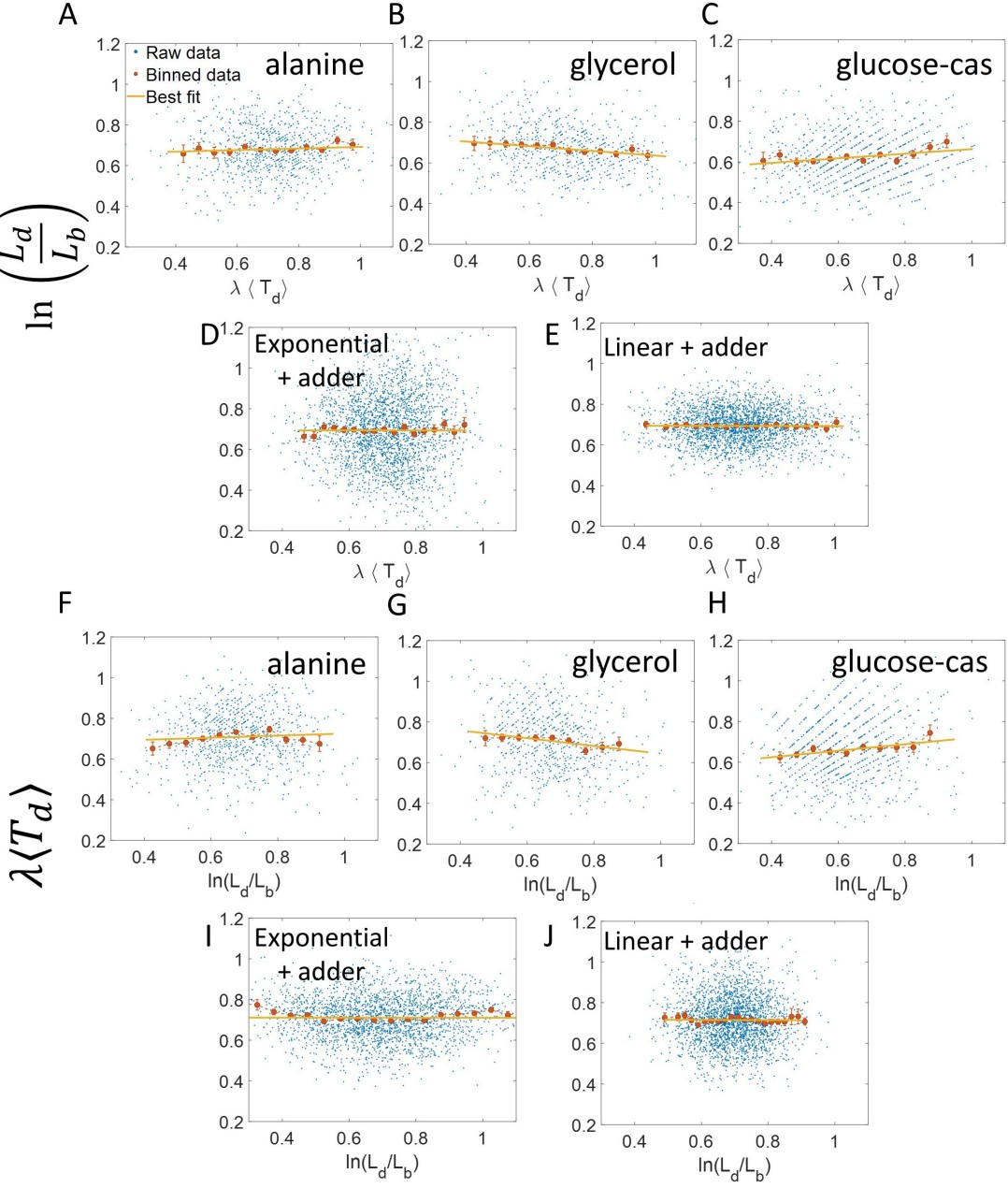

**Appendix 2—figure 1.** $\ln(\frac{L_d}{L_b})$ vs $\langle T_d \rangle \lambda$ and its flipped axes plots. (**A-E**) $\ln(\frac{L_d}{L_b})$ vs $\langle T_d \rangle \lambda$ are shown for A. Experimental data in alanine medium. B. Experimental data in glycerol medium. C. Experimental data in glucose-cas medium. D. Simulations of the adder model where cells grow exponentially, carried out for N = 2500 cells. (**E**) Simulations of the adder model where cells grow linearly, carried out for N = 2500 cells. F-J. For the same order of the above experimental conditions and simulations, $\langle T_d \rangle \lambda$ vs $\ln(\frac{L_d}{L_b})$ plots are shown. In all of the plots, blue represents the raw data, red represents the binned data, and the yellow line represents the best linear fit obtained by applying linear regression on the raw data. In all of the plots, the slope of the best linear fit is close to zero. Thus, we find that these plots are not a suitable method to differentiate between linear and exponential growth as they provide a similar best linear fit.

**Appendix 2—table 1.** The slope and the intercept of the best linear fit along with their 95 % confidence intervals (CI) obtained on performing linear regression on experimental data. The data is collected for cells growing in M9 alanine, glycerol and glucose-cas media (*Tiruvadi-*

*Krishnan et al., 2021*).

| Media | No. of | $T_d$ | $\ln(\frac{L_d}{L_b})$ vs $\langle T_d \rangle \lambda$ plot | | $\langle T_d \rangle \lambda$ vs $\ln(\frac{L_d}{L_b})$ plot | |
|---|---|---|---|---|---|---|
| | | | Slope (with 95% CI) | Intercept (with 95% CI) | Slope (with 95% CI) | Intercept (with 95% CI) |
| Alanine | 816 | 214 | 0.04 (-0.01, 0.09) | 0.65 (0.62, 0.69) | 0.05 (-0.01, 0.12) | 0.67 (0.63, 0.72) |
| Glycerol | 648 | 164 | –0.12 (-0.16,–0.07) | 0.75 (0.71, 0.79) | –0.19 (-0.27,–0.11) | 0.83 (0.78, 0.89) |
| Glucose-cas | 737 | 65 | 0.11 (0.06, 0.16) | 0.55 (0.52, 0.58) | 0.16 (0.09, 0.23) | 0.56 (0.51, 0.61) |

