## [Editor Report]

In this manuscript, the authors describe a generative model-based framework to better analyze stochastic growth data, including bacterial cell growth. They show how this framework can be applied to gain insight into the processes underlying these phenomena. This work is well-supported by simulations and data analysis and will likely be of interest to those trying to understand the processes governing bacterial growth, as well as those studying stochastic growth processes in biology more broadly.

---

## [Decision Letter]

**Decision letter after peer review:**

Thank you for submitting your article "To bin or not to bin: analyzing single-cell growth data" for consideration by *eLife*. Your article has been reviewed by 3 peer reviewers, one of whom is a member of our Board of Reviewing Editors, and the evaluation has been overseen by Aleksandra Walczak as the Senior Editor. The following individual involved in review of your submission has agreed to reveal their identity: Benjamin P Bratton (Reviewer #3).

Essential revisions:

1) The final result (Figure 4) is somewhat disconnected from the majority of the paper that precedes it. Specifically, the authors' procedure that resolves exponential vs. non-exponential growth results in *E. coli* in alanine being deemed exponential (Figure 2B) only to later be revealed as non-exponential (Figure 4A), albeit weakly. Furthermore, the procedure advertised as distinguishing exponential from linear growth (Figure 3B), when applied to the data, reveals neither (Figure 4). This makes the main point of the paper (the demonstration and resolution of pitfalls) feel disconnected from its application to a particular case, which is more nuanced and likely leaves many questions unanswered.

2) The title ("To bin or not to bin…") implies that binning is the main culprit behind potentially misleading analyses, but one of the reviewers argued that in the end, it is linear regression. Each of the two main pitfalls and their resolution would be unchanged if the data were never binned. Binning affects the apparent curvature of the y vs x relationship, but this reads as a more minor point. Therefore, the title may be a bit misleading in service of its poeticism.

3) The authors look at different choices of binning dimensions but do not sufficiently explore the power of their generative model to perform (un)weighted regression or parameter estimation from the not-binned data. They do explore the unbinned data from an analytical statistics approach in sections 5.4.1 and 5.5 but this is not yet extensively explored in the figures and/or discussion.

4) This manuscript could be improved by including a discussion on binning single-cell length trajectories, those taken on a single cell within one division cycle, either based on the length or on time. Experimentally, my understanding is that most growth measurements are done on a fixed \Δ t basis, and therefore measurements are averaged using length as a function of time. Another option, one that I have not seen used, is to use a \Δ L basis and measure the time to grow either a specific amount or a specific fractional amount. Of course, these are different ways to slice the same cake, and such an extension may be useful to expand the authors' argument that one needs to utilize a generative model to be able to assess their choice in statistical analysis.

5) As the formulation of this approach borrows from and is built heavily on previous data and previous statistical frameworks, it would be helpful to have more discussion on what is new to this particular manuscript. This is always a difficult balance as one needs the context of the previous work to understand the symbology used, but just reiterating what is already in the literature can lead to a dilution of the new/novel/important discoveries of the manuscript.

6) The authors should resolve the use of the term "cell size" to mean both "cell length" and "cell volume" (line 107, line 431/432). The authors briefly touch on a related concept of length growth vs biomass growth in 343-357. As a first, naively simple model of cell growth, it is possible to come up with a simple model for linear vs volumetric growth by imagining that a cell grows exponentially in length from birth to death. In this simple model, at the appropriate time, it switches from growing a cylinder to growing hemispherically endcaps, as it needs to perfectly duplicate its shape. In this simple model, one can easily see that the instantaneous, relative volumetric and length growth rates are not identical; vary throughout the cell cycle; and even their average over the whole cell cycle is not 1. While the authors mention that the average diameter is quite robust between growth conditions, only in the limit that the cell is very long and not a spherocylinder does the approximation that shape does not matter come in to play. Granted, this naïve model does not incorporate the wealth of information known about division timing or geometric changes during growth, and underscores the authors' main point that a specific model of the dynamical process should be included.*Reviewer #1:*

In this manuscript, the authors describe a generative model-based framework to better analyze stochastic growth data, including bacterial cell growth. They show how this framework can be applied to gain insight into the processes underlying these phenomena. More specifically, they start by showing how binning along different axes in the

This work is well-supported by simulations and data analysis and will likely be of interest to those trying to understand the processes governing bacterial growth, as well as those studying stochastic growth processes in biology more broadly.

Strengths:

– The choice and execution of the simulations were sensible and well-done, respectively, and they provided clarity as to the overall message of the manuscript.

– The conclusions are well-supported by the data.

– I found the writing to be clear throughout.

Weaknesses:

– It would be good to have a more extensive discussion about what is specifically new here. This is not my particular field, so having a bit more of an introduction about methods beyond binning (if any) that have emerged to understand these data.*Reviewer #2:*

The manuscript by Kar et al., uses single-cell experiments, simulations, and theory to investigate a common method of determining whether cell size grows exponentially. Specifically, they show that a relationship that should adhere to y = x for exponential growth on average (where x is the product of the division time and mean growth rate, and y is the log of the ratio of the birth and division size), in fact deviates from y = x with noise, because x contains noise that y does not (growth rate noise). This makes exponential growth seem non-exponential. The resolution, they show, is to plot x vs. y instead. Additionally, they show that when plotting x vs. y for linear growth, the relationship is coincidentally very close to x = y. This makes linear growth seem exponential. The resolution, they show, is to plot the instantaneous growth rate vs. the normalized cell age, which will decrease for linear growth and will be constant for exponential growth. Applying this protocol to *E. coli* size data, they find that the growth rate actually increases weakly with age, indicating somewhat superexponential growth.

Strengths

The insights in this manuscript are highly important for the field to know. The fact that exponential growth can masquerade as non-exponential growth and vice versa means that much confusion likely exists in a field that is already surprisingly complex given how simple its questions are to state. The fact that the authors offer resolutions to these pitfalls means that this work should add clarity to the field and move it forward in a meaningful way.

The conclusions are well supported by the data. For example, in the case of seemingly non-exponential growth, the problem is presented by the experimental data, reproduced by the simulations, and explained by the theory, while the resolution is inspired by the theory, proven by the simulations, and demonstrated by the experimental data.

The manuscript is well written. While it is rooted in careful analysis, it remains understandable to a largely non-quantitative audience.

Weaknesses

The final result (Figure 4) is somewhat disconnected from the majority of the paper that precedes it. Specifically, the authors' procedure that resolves exponential vs. non-exponential growth results in *E. coli* in alanine being deemed exponential (Figure 2B) only to later be revealed as non-exponential (Figure 4A), albeit weakly. Furthermore, the procedure advertised as distinguishing exponential from linear growth (Figure 3B), when applied to the data, reveals neither (Figure 4). This makes the main point of the paper (the demonstration and resolution of pitfalls) feel disconnected from its application to a particular case, which is more nuanced and likely leaves many questions unanswered.

The title ("To bin or not to bin…") implies that binning is the main culprit behind potentially misleading analysis, but I would argue that in the end, it is linear regression. Each of the two main pitfalls and their resolution would be unchanged if the data were never binned, I believe. Binning affects the apparent curvature of the y vs x relationship, but this reads as a more minor point. Therefore, the title may be a bit misleading in service of its poeticism.*Reviewer #3:*

Kar et al., examine an interesting and important question of how to make sense of large sets of observational data, specifically cell length data, which may or may not be consistent with various underlying biological mechanisms. As datasets improve in their technical quality (increasing spatiotemporal resolution, increasing numbers of observations), there is hope that the community will be able to resolve differences between underlying cell biological mechanisms of cell size homeostasis. As the authors point out, these interpretations and analyses require statistical analysis that can accurately perform the model selection or parameter estimation task of interest.

1) The authors succeed in bringing attention to the issue of appropriate binning when analyzing large datasets. The authors focus their figures and discussion on an important, and practical issue, as many researchers perform linear regression on binned data. The title and framing of the manuscript imply that it will provide a comparison with statistical methods that do not involve binning. The authors look at different choices of binning dimensions, but do not sufficiently explore the power of their generative model to perform (un)weighted regression or parameter estimation from the not-binned data. They do explore the unbinned data from an analytical statistics approach in section 5.4.1 and 5.5 but this not yet extensively explored in the figures and/or discussion.

2) The authors succeed in walking the reader through the power of examining a specific mechanism/model and the statistical properties of that model. In this case, the authors do this with both a model of cell length homeostasis that comes from exponential growth or linear growth with homeostatic feedback. This approach rests heavily on previous work from the same group including a reframing of the statistical correlations between different observables that was explored in the included references [13] and [16]. This reframing is complemented by additional experiments and reanalysis of various published experimental datasets.

---

## [Author Response]

Essential revisions:1) The final result (Figure 4) is somewhat disconnected from the majority of the paper that precedes it. Specifically, the authors' procedure that resolves exponential vs. non-exponential growth results in *E. coli* in alanine being deemed exponential (Figure 2B) only to later be revealed as non-exponential (Figure 4A), albeit weakly. Furthermore, the procedure advertised as distinguishing exponential from linear growth (Figure 3B), when applied to the data, reveals neither (Figure 4). This makes the main point of the paper (the demonstration and resolution of pitfalls) feel disconnected from its application to a particular case, which is more nuanced and likely leaves many questions unanswered.

We try to bridge the gap between Figure 4 and the text preceding it by using simulations of super-exponential growth as mentioned in response to Comment 1 of Reviewer 2.

2) The title ("To bin or not to bin…") implies that binning is the main culprit behind potentially misleading analyses, but one of the reviewers argued that in the end, it is linear regression. Each of the two main pitfalls and their resolution would be unchanged if the data were never binned. Binning affects the apparent curvature of the y vs x relationship, but this reads as a more minor point. Therefore, the title may be a bit misleading in service of its poeticism.

We have changed the title of the article to a more apt one – ”Distinguishing different modes of growth using single-cell data”.

3) The authors look at different choices of binning dimensions but do not sufficiently explore the power of their generative model to perform (un)weighted regression or parameter estimation from the not-binned data. They do explore the unbinned data from an analytical statistics approach in sections 5.4.1 and 5.5 but this is not yet extensively explored in the figures and/or discussion.

We apologise for the confusion and we try to clarify it in the manuscript as stated previously in response to Comment 1 of Reviewer 3.

4) This manuscript could be improved by including a discussion on binning single-cell length trajectories, those taken on a single cell within one division cycle, either based on the length or on time. Experimentally, my understanding is that most growth measurements are done on a fixed \Δ t basis, and therefore measurements are averaged using length as a function of time. Another option, one that I have not seen used, is to use a \Δ L basis and measure the time to grow either a specific amount or a specific fractional amount. Of course, these are different ways to slice the same cake, and such an extension may be useful to expand the authors' argument that one needs to utilize a generative model to be able to assess their choice in statistical analysis.

In Section 3 of the manuscript, we discuss binning in relation to the growth rate of single-cell length trajectories as a function of age, and time, and the elongation speed as a function of size. In the simulations conducted in Section 3, we consider the cell lengths measurements to be at equal intervals of time within the cell cycle so as to mimic our mother machine experiments. However, using our methodology of analysing based on a model, it is possible to speculate on the outcome of the data analysis methods in the case of mother machine experiments where measurements are done at equal intervals of length (∆*L*).

We perform simulations in which lengths within the cell cycle are recorded at ∆*L* = 0.05*_µ_m* intervals and the time from birth corresponding to each length increase is also noted. The results from the simulation are shown in Author response image 1. We find that the binned data trend and the best linear fit for the cells growing exponentially following the adder model deviate from the y=x line for ln (LdLb)vs ⟨λ⟩Td plot (Author response image 1). This is expected from Equations. 2, 27 and 28 of the main text. On interchanging the axes, the best linear fit and the binned data trend are expected to follow the y=x line for exponential growth. This is shown in Author response image 1 for simulations of cells following the adder model and undergoing exponential growth. For cells undergoing linear growth and birth controlling the division, the slope and intercept of the best linear fit of ⟨*λ*⟩*T_d_* vs ln⁡(LdLb) are analytically found to be approximately 1.04 and -0.03 respectively. This is shown in Author response image 1 for simulations of linear growth and cells following the adder model. The binned growth rate trend in growth rate vs age plots are found to be almost constant for cells undergoing exponential growth irrespective of them following the adder model or the adder per origin model (Author response image 1). In the case of linear growth, the growth rate is expected to decrease with cell age. This is observed for simulations of linearly growing cells following the adder model (Author response image 1). We also observe an increase in growth rate in agreement with the simulations of cells following the adder model and undergoing faster than exponential growth (Author response image 1). All of the results presented here are very similar to that of simulations where the cell lengths were measured at equal intervals in time. We also comment about it in p.37 line 685 of the revised manuscript:

“Note that if the measurements were to be carried out at equal length intervals instead of time, the results discussed in the paper would still remain unchanged.”

**Author response image 1. sa2fig1:** Measurements at ∆*L* intervals: Results are shown for simulations carried out for N=2500 cells, where the cell length measurements are done at ∆*L* = 0. 05_µ_m intervals of length instead of equal intervals in time. For exponentially growing cells following the adder model, (A) ln⁡(LdLb)vs<λ>Td plot is shown. The binned data trend (red) and the best linear fit (yellow) deviate from the y=x line (black dashed line). (B) <λ>Tdvsln⁡(LdLb) plot is shown. The binned data trend (red) and the best linear fit (yellow) are close to the y=x dependence (black dashed line). C. For linearly growing cells following the adder model, the binned data trend (red) and the best linear fit (yellow) of the <λ> Tdvsln⁡(LdLb) plot closely follow the y=x dependence (black dashed line). D. The growth rate vs age plots for exponentially growing cells following the adder (purple circles) and adder per origin model (magenta diamonds) are constant while for linear growth (green squares) and super-exponential growth (red triangles) following adder model, the growth rate decreases and increases respectively in agreement with the underlying mode of growth. In summary, the results obtained for ∆*L* measurements are similar to that obtained for ∆*t* measurements.

5) As the formulation of this approach borrows from and is built heavily on previous data and previous statistical frameworks, it would be helpful to have more discussion on what is new to this particular manuscript. This is always a difficult balance as one needs the context of the previous work to understand the symbology used, but just reiterating what is already in the literature can lead to a dilution of the new/novel/important discoveries of the manuscript.

We try to be clearer in the manuscript, as detailed in the response to Comment 1 of Reviewer 1.

6) The authors should resolve the use of the term "cell size" to mean both "cell length" and "cell volume" (line 107, line 431/432). The authors briefly touch on a related concept of length growth vs biomass growth in 343-357. As a first, naively simple model of cell growth, it is possible to come up with a simple model for linear vs volumetric growth by imagining that a cell grows exponentially in length from birth to death. In this simple model, at the appropriate time, it switches from growing a cylinder to growing hemispherically endcaps, as it needs to perfectly duplicate its shape. In this simple model, one can easily see that the instantaneous, relative volumetric and length growth rates are not identical; vary throughout the cell cycle; and even their average over the whole cell cycle is not 1. While the authors mention that the average diameter is quite robust between growth conditions, only in the limit that the cell is very long and not a spherocylinder does the approximation that shape does not matter come in to play. Granted, this naïve model does not incorporate the wealth of information known about division timing or geometric changes during growth, and underscores the authors' main point that a specific model of the dynamical process should be included.

Thank you for raising this point. In the revised version, we point out that cell length is used as a measure of cell size. Thus, the growth rate trend shown in the manuscript is the length growth rate.

However, other cell geometry parameters such as surface area and volume can be used to represent cell size and their respective growth rates might differ from the length growth rate. Theoretically, a comparison between the different rates requires choosing a specific model of cell growth as the reviewers point out. In the revised version of the manuscript, we try to compare the different rates assuming a simple cell morphology and using multiple models of cell growth. We add a discussion of these models as Appendix 1 in the revised manuscript:

“In the paper, we use cell length to represent cell size. However, other cell size characteristics such as cell surface area and cell volume could also be used to denote cell size. […] The width of the cells is assumed to be 0.35 *µm*. We find that before constriction, the length growth rate increases to a small extent (≈ 6%). However, after constriction there is a rapid increase in length growth rate. The mode of growth in length and volume are not identical.”

In conclusion, length growth rate and volume growth rate do vary and we try to state it more clearly in p.6 line 116 of the revised text:

“Note that we consider length to reflect cell size in this paper rather than other cell geometry characteristics such as surface area and volume. The length growth rate that we elucidate in the paper can be different from the cell volume growth rate as shown in Appendix 1 assuming a simple cell morphology and exponential growth. Using the same cell morphology, we also find the length growth rate to be identical to cell surface growth rate.”

We also reiterate this in Section 5, p.20 line 472 of the revised text:

“To reiterate, the length growth is not the same as cell volume growth as shown in Appendix 1.”

Reviewer #1:In this manuscript, the authors describe a generative model-based framework to better analyze stochastic growth data, including bacterial cell growth. They show how this framework can be applied to gain insight into the processes underlying these phenomena. More specifically, they start by showing how binning along different axes in theThis work is well-supported by simulations and data analysis and will likely be of interest to those trying to understand the processes governing bacterial growth, as well as those studying stochastic growth processes in biology more broadly.

We thank Reviewer 1 for appreciating the methods used in the work and its scope.

Strengths:– The choice and execution of the simulations were sensible and well-done, respectively, and they provided clarity as to the overall message of the manuscript.– The conclusions are well-supported by the data.– I found the writing to be clear throughout.Weaknesses:– It would be good to have a more extensive discussion about what is specifically new here. This is not my particular field, so having a bit more of an introduction about methods beyond binning (if any) that have emerged to understand these data.

Binning and linear regression are the most commonly used methods for probing the correlations between variables obtained using single cell data analysis [Ho *et al.* (2018), Jun *et al.* (2018)]. In this paper, we try to address the pitfalls associated with applying these simple procedures. In the revised version, we put more emphasis on linear regression throughout the manuscript for example in the ”Introduction” (p.4 line 54 of the revised text):

“While binning may provide a smooth non-linear relation between variables, linear regression is used to find a linear relationship between the variables. In addition to binning, we use the ordinary least squares regression where the slope and the intercept of the best linear fit line are obtained by minimizing the squared sum of the difference between the dependent variable raw data and the predicted value. Here, the best fit/the best linear fit is obtained using the raw data and not the binned data. Similar to binning, the assumption underlying linear regression is that our knowledge of x-axis variable is precise while the noise is in the y-axis variable.”

Following this comment, we added the best linear fits to the Figures 2C, 2D and 3A.

We also try to clearly point out the sections which are not novel to the paper. For example, the adder model in Figure 1A has been discussed previously. We emphasize that in p.5 line 77 of the revised text:

“This previously discussed example demonstrates and reiterates the use of statistical analysis on single-cell data to understand the underlying cell regulation mechanisms.”

Similarly, the novel results of the paper such as obtaining the best linear fits for the plots ln⁡(LdLb) vs ⟨*λ*⟩*T_d_* and its flipped axes are based on a class of models studied by Eun *et al.* (2018). We try to make it clearer in p.7 line 133 of the revised manuscript:

“For that purpose, we use a previously studied model [Eun *et al.* (2018)] which considers growth to be exponential with the growth rate distributed normally and independently between cell cycles with mean growth rate ⟨*λ*⟩ and standard deviation *CV_λ_*⟨*λ*⟩.”

Reviewer #2:[…]WeaknessesThe final result (Figure 4) is somewhat disconnected from the majority of the paper that precedes it. Specifically, the authors' procedure that resolves exponential vs. non-exponential growth results in *E. coli* in alanine being deemed exponential (Figure 2B) only to later be revealed as non-exponential (Figure 4A), albeit weakly. Furthermore, the procedure advertised as distinguishing exponential from linear growth (Figure 3B), when applied to the data, reveals neither (Figure 4). This makes the main point of the paper (the demonstration and resolution of pitfalls) feel disconnected from its application to a particular case, which is more nuanced and likely leaves many questions unanswered.

We attempt to resolve the gap between Figure 4 and the text preceding it by showing that growth rate vs age plot can be used to infer the modes of growth, including, but not limited to, exponential and linear growth. To verify this, we simulate the adder model for cells undergoing super-exponential growth. The binned growth rate trend as a function of age for super-exponential growth is shown in Figure 3B and the following text is added to p.13 line 270 of the revised manuscript:

“Thus, the two growth modes (exponential and linear) could be differentiated using the growth rate vs age plot (for details see Section 5.7). However, the growth rate vs age plots can be used to infer the mode of growth beyond the two discussed above. We show this by using simulations of cells following the adder model and undergoing faster than exponential or super-exponential growth (see the Simulations section for details). In such a case, the growth rate is expected to increase. This increase in growth rate is shown in Figure 3B using simulations. The binned data trend (red triangles) again matches the growth rate mode used in the simulations (red dotted line). Thus, the growth rate vs age plots are a consistent method to distinguish linear from exponential and super-exponential growths.”

The details about the simulation of super-exponential growth has been added to the Simulations section.

We have also added theoretical predictions (dotted lines of same color) to the growth rate vs age curves of different models shown in Figure 3B of the revised manuscript, that agree well with our simulations.

The title ("To bin or not to bin…") implies that binning is the main culprit behind potentially misleading analysis, but I would argue that in the end, it is linear regression. Each of the two main pitfalls and their resolution would be unchanged if the data were never binned, I believe. Binning affects the apparent curvature of the y vs x relationship, but this reads as a more minor point. Therefore, the title may be a bit misleading in service of its poeticism.

We thank the reviewer for the comment and we have changed the title to a more apt one for the manuscript- ”Distinguishing different modes of growth using single-cell data”.

Reviewer #3:Kar et al., examine an interesting and important question of how to make sense of large sets of observational data, specifically cell length data, which may or may not be consistent with various underlying biological mechanisms. As datasets improve in their technical quality (increasing spatiotemporal resolution, increasing numbers of observations), there is hope that the community will be able to resolve differences between underlying cell biological mechanisms of cell size homeostasis. As the authors point out, these interpretations and analyses require statistical analysis that can accurately perform the model selection or parameter estimation task of interest.

We thank Reviewer 3 for the comments. Indeed, our message in the paper is to use underlying biological models to aid the inference of biological mechanisms using various statistical analyses methods.

1) The authors succeed in bringing attention to the issue of appropriate binning when analyzing large datasets. The authors focus their figures and discussion on an important, and practical issue, as many researchers perform linear regression on binned data. The title and framing of the manuscript imply that it will provide a comparison with statistical methods that do not involve binning. The authors look at different choices of binning dimensions, but do not sufficiently explore the power of their generative model to perform (un)weighted regression or parameter estimation from the not-binned data. They do explore the unbinned data from an analytical statistics approach in section 5.4.1 and 5.5 but this not yet extensively explored in the figures and/or discussion.

We apologize for the confusion. Throughout the paper, we perform linear regression on the raw data and not the binned data. We have now added a clarification statement in p.4 line 58 of the revised text:

“Here, the best fit/the best linear fit is obtained using the raw data and not the binned data.”

Further, we have changed the title of the manuscript to ”Distinguishing different modes of growth using single-cell data”. The paper aims to bring forth the issues in both binning and linear regression which arise from similar sources i.e., the intrinsic noise affecting the x-axis variable and the inspection bias. We discuss these issues in relation to mode of growth in single cells as the new title now states.

In addition to section 5.4.1 and 5.5 where we discuss calculating the best fit line for exponential and linear growth, we also try to explicitly mention linear regression on non-binned data in the figures, and the Results and Discussion section. This is shown by the addition of the best linear fit/best fit in Figures 2C, 2D and 3A.